# Petrogenesis of Ultramafic Lamprophyres from the Terina Complex (Chadobets Upland, Russia): Mineralogy and Melt Inclusion Composition

**Ilya Prokopyev** [1,2,]*[ⓘ]**, Anastasiya Starikova** [1,2][ⓘ]**, Anna Doroshkevich** [1,3][ⓘ]**, Yazgul Nugumanova** [1,2] **and Vladislav Potapov** [1,2]

[1]  Sobolev Institute of Geology and Mineralogy, Siberian Branch of the Russian Academy of Sciences, Akademika Koptyuga Avenue 3, 630090 Novosibirsk, Russia; a_sklr@mail.ru (A.S.); doroshkevich@igm.nsc.ru (A.D.); nugumanovayn@igm.nsc.ru (Y.N.); vladislavpotapovjobmail@yandex.ru (V.P.)

[2]  Department of Geology and Geophysics, Novosibirsk State University, Pirogova Street 1, 630090 Novosibirsk, Russia

[3]  Geological Institute, Siberian Branch of the Russian Academy of Sciences, Sakhyanova Street 6a, 670047 Ulan-Ude, Russia

*  Correspondence: prokop@igm.nsc.ru; Tel.: +7-(383)-373-0526-(719)

**Abstract:** The mineral composition and melt inclusions of ultramafic lamprophyres of the Terina complex were investigated. The rocks identified were aillikites, mela-aillikites, and damtjernites, and they were originally composed of olivine macrocrysts and phenocrysts, as well as phlogopite phenocrysts in carbonate groundmass, containing phlogopite, clinopyroxene and feldspars. Minor and accessory minerals were fluorapatite, ilmenite, rutile, titanite, and sulphides. Secondary minerals identified were quartz, calcite, dolomite, serpentine, chlorite, rutile, barite, synchysite-(Ce), and monazite-(Ce). Phlogopite, calcite, clinopyroxene, Ca-amphibole, fluorapatite, magnetite, and ilmenite occurred as daughter-phases in melt inclusions. The melt inclusions also contained Fe–Ni sulphides, synchysite-(Ce) and, probably, anhydrite. The olivine macrocrysts included orthopyroxene and ilmenite, and the olivine phenocrysts included Cr-spinel and Ti-magnetite inclusions. Crystal-fluid inclusions in fluorapatite from damtjernites contain calcite, clinopyroxene, dolomite, and barite. The data that were obtained confirm that the ultramafic lamprophyres of the Terina complex crystallized from peridotite mantle-derived carbonated melts and they have not undergone significant fractional crystallization. The investigated rocks are considered to be representative of melts that are derived from carbonate-rich mantle beneath the Siberian craton.

**Keywords:** ultramafic lamprophyre; aillikite; damtjernite; melt inclusion; Siberian Craton

## 1. Introduction

The term "aillikite" was first introduced in 1939 to describe the carbonate-rich varieties of lamprophyres from the Aillik Bay area of Labrador. These differ from alnöites, due to their lack of melilite [1]. In 1986, the terms melilitic lamprophyres (i.e., alnöites) and melilite-free carbonate-rich and feldspar-/foid-bearing varieties, such as aillikite and damtjernite/ouachitite, respectively, were originally suggested [2], but they were not accepted by the International Union of Geological Sciences (IUGS) subcommittee on classification [3,4]. Eventually, the end members of the ultramafic lamprophyres (UMLs) were integrated into the IUGS Classification of Igneous Rocks by Tappe et al. [5], and they are described below. The melilite-bearing rocks now have a separate classification in the IUGS.

Aillikites are carbonate-rich UMLs, which are composed of olivine and phlogopite macrocrysts and/or phenocrysts in a groundmass of primary carbonate, phlogopite, spinel, ilmenite, rutile, perovskite, Ti-rich garnet, and apatite [5,6]. Mela-aillikites have a color index >90% because of the presence of clinopyroxene and/or richterite in the groundmass (instead of carbonate) [5]. Monticellite may occur as rare inclusions. Damtjernites are feldspathoid- and/or alkali feldspar containing UML, consisting of olivine, phlogopite, and clinopyroxene macrocrysts and/or phenocrysts, and the groundmass is composed of phlogopite/biotite, clinopyroxene, spinel, ilmenite, rutile, perovskite, Ti-rich garnet, titanite, apatite, and primary carbonate, with essential minor quantities of nepheline and/or alkali feldspar [5]. Petrological investigations of the ultramafic lamprophyres and carbonatites of the Chuktukon complex show that the rocks were formed from the primary melts that were derived from a moderately depleted mantle [7]. Low-degree partial melting of a phlogopite-carbonate metasome garnet peridotite under the influence of the Siberian plume heat produced the primary melts [7]. Melt inclusion investigations in zircon of the Chuktukon complex show the presence of Na–K- and Ba–Sr–REE–Ca-carbonates, as well as alkali silicate and phosphate (fluorapatite) phases in the inclusions [8]. The trace element composition of olivines from the Chadobets alkaline complex confirms that the primary melts were derived from a carbonate-rich source [9], and this can be traced in the Sr–Nd–Pb–C–O isotope characteristics of the Chadobets UML rocks [10].

In this paper, we present new data regarding the mineralogy of the UML rocks of the Terina complex and show the first data on the parental melt composition for these rocks based on melt inclusion studies.

## 2. Geological Setting

The Chadobets alkaline complex is located in the southwestern part of the Siberian Craton (Figure 1a). It contains melilitites, ultramafic lamprophyres (ailllikites, mela-aillikites and damtjernites), and carbonatites, which form the different phases of intrusion within the Chuktukon and Terina complexes (Figure 1b). Regarding the age of the Chadobets alkaline rocks, perovskite from the Chuktukon complex UMLs is 252 ± 12 Ma (U-Pb, SHRIMP II) and rippite (a newly discovered mineral $K_2(Nb,Ti)_2(Si_4O_{12})O(O,F)$) from carbonatites is 231 ± 2.7 Ma (Ar–Ar) [7,11,12]. The reported ages of formation of aillikites are 243.4 ± 3.1 and 241 ± 1 Ma (Ar–Ar, phlogopite, and Rb–Sr, respectively) [10]. The emplacement of the Chadobets alkaline rocks was coeval with Siberian plume activity and with the formation of the Siberian large igneous province (LIP) (Figure 1a) [7], containing flood basalts [13], carbonatites, and UMLs of the Maymecha–Kotuy province [14–17], as well as with Siberian kimberlite from Anabar and Olenek region, and lamproite intrusions [18–24].

The Chadobets UML–carbonatite complex is confined to the Chadobets upland about 40 × 50 km, and the structure of the upland is elongated in the NS direction (Figure 1b). The formation of the Chadobets upland is linked to the ascent diapir, and the geophysical data reveal an intermediate magma chamber at a depth of about 4–8 km [25]. The Chadobets upland is situated at the intersection of two Neoproterozoic graben basins, which are located in the Angara-Kotuy large-scale rift system [7,9,10,25–27].

The Chadobets alkaline complex is formed by several phases of alkaline rock intrusions located within the two main ledges, which are usually described as the Terina and Chuktukon complexes (Figure 1b) [7,9,10,25–28]. The Terina and Chuktukon complexes consist of alkaline rock varieties, such as allikites, mela-allikites, picrites, and olivine melilitites, forming the first phase of the alkaline rock intrusions, while the Chuktukon complex is predominantly represented by carbonatites, which are the second phase of the Chadobets complex emplacement [7,25,28–30]. The weathered crust of the carbonatites contains the Chuktukon Nb–REE deposit with estimated total reserves of 39.8 Mt of Nb and 486 Mt of REE [31]. Damtjernites represent the last phase of the Chadobets complex (Figure 1b).

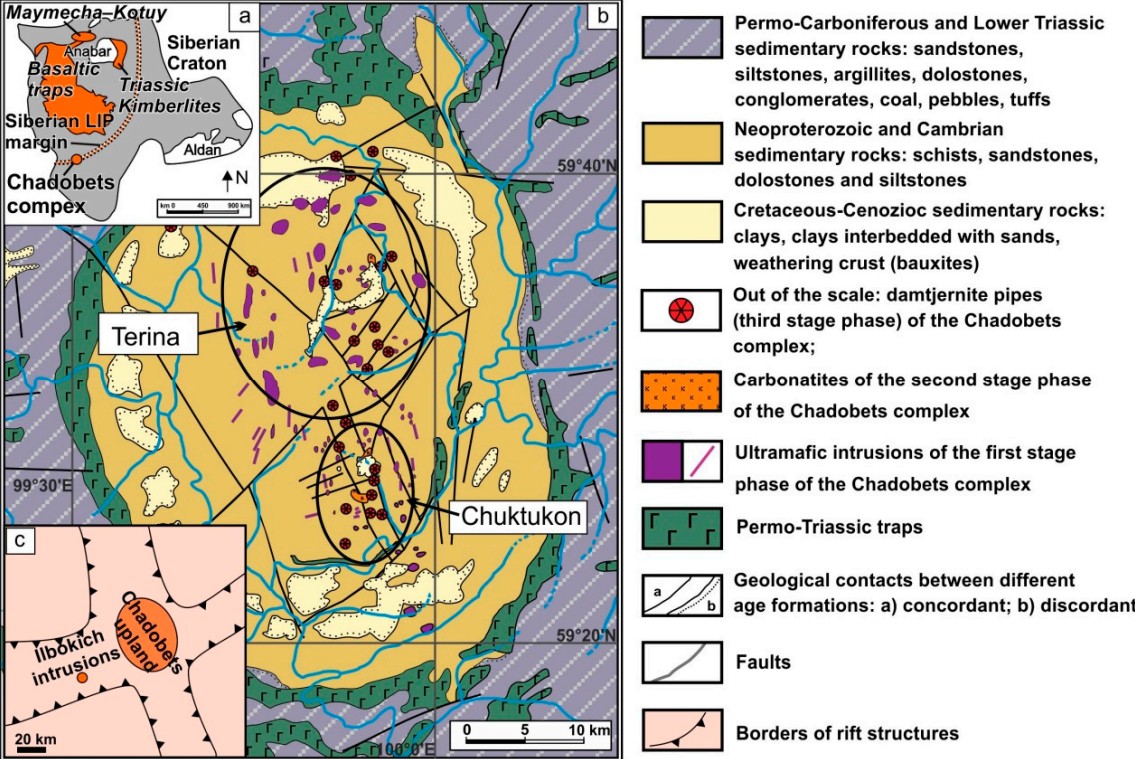

**Figure 1.** (**a**) Location of the Chadobets ultramafic lamprophyre (UML)–carbonatite complex within the Siberian Large Igneous Provence (LIP) on the Siberian Craton [24]. (**b**) Geological scheme of the Chadobets alkaline complex [7,25]. (**c**) Location of the Devonian Ilbokich complex [26].

UMLs of the first phase form dyke swarms and sills (ranging in size from 1–3 up to 120 m) [7,25,28–30]. The alkaline intrusions have sharp contacts with country rocks and they are distinguished by a banded structure. The carbonatite bodies are small stocks (up to 2.5 × 1.5 km), dykes, and sills (20 m × 2–3 km in length) that intrude the UMLs and surrounding rocks. Damtjernite bodies produce pipes and diatremes (several tens of meters in size) and they are usually located in the central parts of the Chuktukon and Terina complexes (Figure 1b). The damtjernites cut the earlier alkaline rocks and usually contain their xenoliths as well as country rock fragments. In general, all of the magmatic varieties were subjected to the intense processes of hydrothermal-metasomatic alterations and weathering. Moreover, the UML-carbonatite bodies are surrounded by thin (several cm) potassic feldspar-aegirine or apatite-phlogopite fenite zones with surrounding rocks [7,25,28–30].

The alkaline intrusions of the Chadobets complex are not exposed at the surface within the upland, but they are covered by Cretaceous–Cenozoic sediments that are made of clays and clays interbedded with sands with weathering crust-forming bauxites (Figure 1b). The rocks surrounding the Chadobets UML-carbonatite complex are Neoproterozoic and Cambrian sedimentary rocks: schists, sandstones, dolostones, and siltstones. The periphery of the Chadobets upland comprises Permian-Carboniferous and Lower Triassic sediments: sandstones, siltstones, argillites, dolostones, conglomerates, coal, pebbles, and tuffs, as well as areas of Permo-Triassic traps (Figure 1b).

The Chadobets upland is situated on the intersection of several rift grabens, and the Devonian Ilbokich UML complex is located approximately 70 km to the southwest of the Chadobets uplift [9,32] (Figure 1c). The age of crystallization of the Ilbokich UML rocks is 399 ± 4 Ma (perovskite, U-Pb) [10]. The aillikite and damtjernite dyke swarms form the Ilbokich UML complex, with individual dykes reaching up to 4–5 m in thickness [9,32]. Up-to-date information regarding the petrology of the Ilbokich and Chadobets UML rocks and their relationship with Siberian plume activity is provided in [7,10,32].

## 3. Sampling Procedure and Analytical Methods

The samples that were used for mineralogical and melt inclusion investigations were collected from the exploration drill cores and the outcrops at the Terina riverbanks. Polished thin-sections were prepared for petrographic investigations and polished rock samples were prepared for ore mineralogy, and these were studied in transmitted and reflected light, respectively, under a polarizing microscope (Olympus BX51) with a photo camera device. The polished rock samples were used to determine the rock textures and mineral assemblage compositions while using energy-dispersive spectrometry in combination with back-scattered electron imaging (BSE) using the TESCAN MIRA 3 LMU JSM-6510LV scanning electron microscope with the energy prefix from X-Max Oxford Instruments for microprobe analysis.

Mineral compositions were determined using a JEOL JXA-8100 electron microprobe (WDS mode, 20 kV, 15 nA, 1–2 μm beam diameter). The accumulation time for analyzing F (using LDE crystal) was 40 s (20 s—counting of background; 20 s—counting of peak for F). The detection limit for F was 477 ppm (0.04 wt%). For mineral analysis, we used a beam current of 10 nA and an acceleration voltage of 15 kV; for Fe–Ti oxides, we used 20 nA and 15 kV; for monazite, we used 40 nA and 20 kV; and, for apatite, we used 10 nA and 20 kV. The peak counting time was 16 s for major elements and 30–60 s for minor elements. For calibration, both natural minerals and synthetic phases were used as standards (element, detection limits in ppm): $SiO_2$ (Si, 158), rutile (Ti, 120), $LiNbO_3$ (Nb, 142), Sr silicate glass (Sr, 442), albite (Na, 176), orthoclase (K, 182), $Al_2O_3$ (Al, 128), F-apatite (Ca, 115; P, 387; F, 477), Mn-garnet (Mn, 129), hematite (Fe, 148), $CePO_4$ (Ce, 236), $LaPO_4$ (La, 272), $BaSO_4$ (S, 178), $NdPO_4$ (Nd, 362), Cl-apatite (Cl, 74), and $PrPO_4$ (Pr, 401).

Major and trace elements of olivines were determined on the JEOL JXA-8100 electron probe microanalyzer. The operation conditions were a beam current of 250 nA and accelerating voltage of 20 kV; the peak count times per element were 15 s for Si, Mg, and Fe; 60 s for Zn and Ca; 90 s for Ni, Co, Al, and Cr; and, 120 s for P and Ti. The data were calibrated against the following standards: olivine Ch1 (Si, Mg, Fe), pyrope (Al), Mn-garnet (Mn), blue diopside (Ca), $Co_3O_4$ (Co), $NiFe_2O_4$ (Ni), $ZnFe_2O_4$ (Zn), $TiO_2$ (Ti), $Cr_2O_3$ (Cr), and fluorapatite (P). The ZAF procedure was applied for matrix correction. The error was less than 0.3 rel.% for major elements; less than 3 rel.% for Ni, Mn, and Ca; 8–15 rel.% for Al, Co, and Ti; 15–30 rel.% for Cr and P; and, 30–40 rel.% for Zn. San Carlos olivine (USNM 111312/444) was used as an internal standard to monitor stability and instrumental drift. Detection limits (3σ) for trace elements were as follows (wt%): 0.001 (CaO), 0.002 ($Al_2O_3$, $P_2O_5$, MnO), 0.003 (NiO, CoO, $TiO_2$), and 0.005 ($Cr_2O_3$, ZnO).

The double-polished thin sections were prepared for melt and fluid inclusion investigations. Raman spectroscopy was applied to determine the composition of the crystalline phases of the inclusions. Raman spectra were obtained on a LabRam HR800 Horiba Jobin Yvon spectrometer, equipped with an optical microscope (Olympus BX41). The 514.5 nm $Ar^+$ laser line was used for spectra excitation. The well-known RRUFF (http://rruff.info) database was used to identify the solid phases. The polished preparations with open melt and fluid inclusions were also used for inclusion composition investigations under the scanning electron microscope.

The investigations were carried out at the Analytical Center for Multi-Elemental and Isotope Research Siberian Branch, Russian Academy of Science (Novosibirsk, Russia).

## 4. Results of Investigations

### 4.1. Petrography and Mineralogy of the UML Rocks

The petrographic and mineralogical characteristics of the alkaline rocks of the Chadobets complex were previously described [7,9,10,25,28–30]. We provide new details regarding the mineral composition of the Terina complex UMLs that has not been presented before.

*Allikites* and *mela-allikites* have massive porphyritic structures (Figure 2a–i). The rocks contain macrocrysts of olivine (20–30 vol%) and phenocrysts of phlogopite (10–20%). The macrocrysts

occur in an interstitial fine-grained groundmass of carbonate (dolomite and calcite) (up to 40%), phlogopite (15–25%), spinel-group minerals, perovskite (15–25%), and clinopyroxene (up to 10–15% for mela-aillikites) (Figure 2h,i). The appearance of feldspar and/or alkali feldspar crystals (up to 10 vol%) in the groundmass makes it possible to attribute the rock to *damtjernites* [5] (Figure 2m–p).

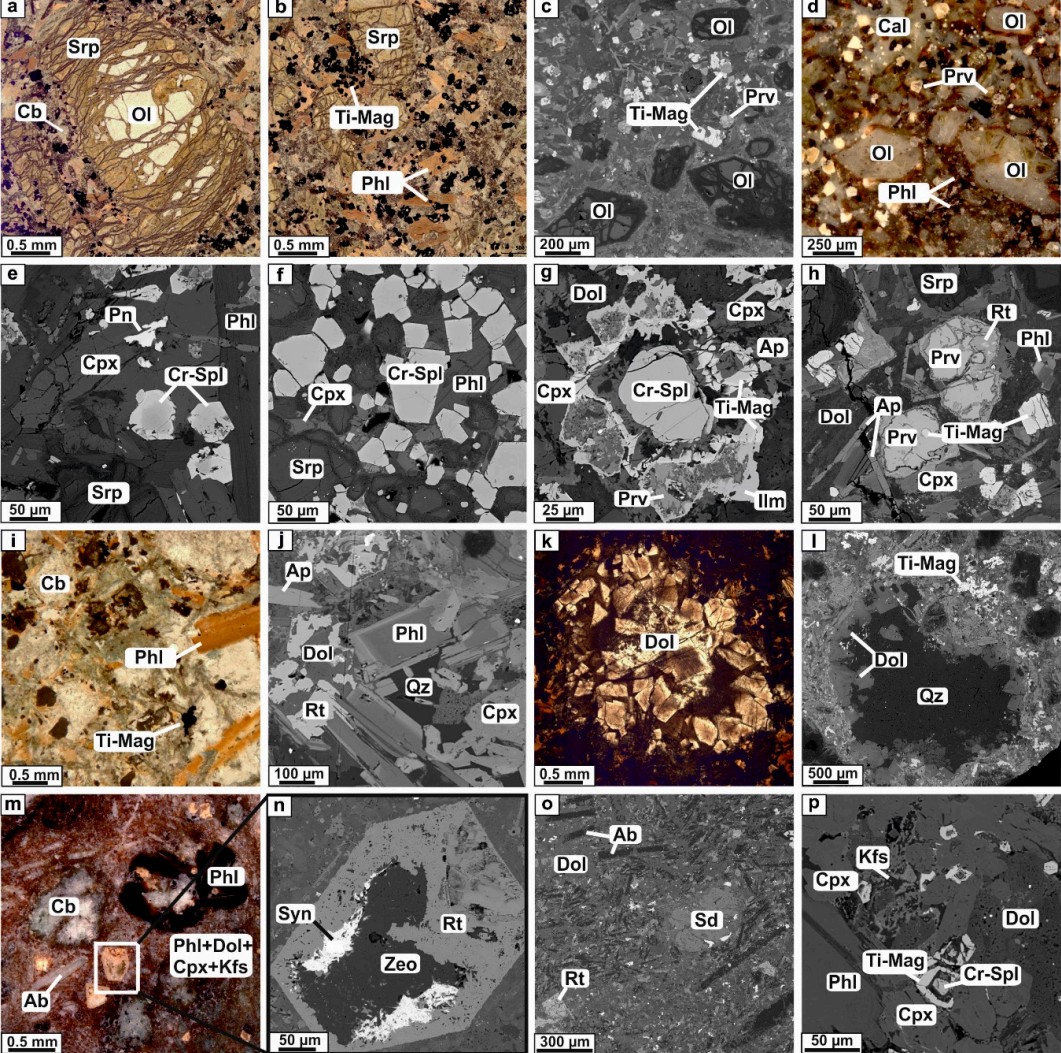

**Figure 2.** Photomicrographs (**a,b,d,i,k,m**) and back-scattered electron imaging (BSE) images (**c,e–h,j,l,n–p**) of mineral assemblages of the Terina complex UMLs: (**a–d**) aillikites, (**e–l**) mela-aillikites, (**m–p**) damtjernites. (**a–d**) Porphyritic structures of aillikites: olivine (Ol) (replaced by serpentine (Srp)) and phlogopite (Phl) macrocrysts are located in the groundmass presented by calcite (Cal) and phlogopite phenocrysts, with Ti-magnetite (Ti-Mag) and perovskite (Prv) mineral assemblages. (**e**) Zonal crystals of the spinel-group minerals with Cr-spinel (Cr-Spl) core and Ti–magnetite rims; the mela-aillikites (**e–l**) differ from the aillikites by the presence of xenomorphic grains of clinopyroxene (Cpx) in the groundmass; (**g,h**) perovskite crystals are replaced by ilmenite (Ilm) and rutile (Rt) with Ti-magnetite association in the carbonate (Cb) matrix; (**i,j**) mineral assembleges of the mela-aillikites; the minor and accessory minerals also include fluorapatite (Ap) and sulphides (ex. pentlandite (Pn)). (**k,l**) The secondary hydrothermal quartz-dolomitic aggregates (globules). (**m–p**) Damtjernite differs from the mela-aillikites by the presence of albite and K-feldspar in the groundmass of the ultramafic lamprophyres. (**m,n**) Perovskite is fully replaced by rutile; zeolite group minerals (Zeol) with synchysite-(Ce) (Syn) trace the hydrothermal processes as well as the formation of siderite (Sd) and rutile aggregates in the groundmass mineral assembleges (**o,p**).

Minor and accessory minerals of the UMLs are fluorapatite, sulphides (pyrite, chalcopyrite, and pentlandite), ilmenite, rutile, and titanite. Secondary minerals include quartz, calcite, dolomite, serpentine, epidote, chlorite, rutile, barite, synchysite-(Ce), and monazite-(Ce) (Figure 2).

The *spinel group minerals* of the Terina UML rocks occur as composite subhedral zonal crystals. The central parts present Cr-spinel with the Ti-magnetite rims (Figure 2e–h,p). The Ti-magnetite crystals consist of up to 7.43 wt% MgO, 0.96–3.67 wt% $Al_2O_3$, 11.08–13.49 wt% $TiO_2$, up to 0.56 wt% MnO, up to 0.49 wt% $V_2O_3$, and 73.15–83.81 wt% $FeO_t$ (Table S1). The spinel aggregates are 25–50 μm in size and they are located in the groundmass. The spinel contains 1.41–30.99 wt% $Cr_2O_3$, 44.05–67.43 wt% FeOt, 4.91–9.56 wt% $Al_2O_3$, and 6.02–16.88 wt% $TiO_2$ (Table S1). The spinel-group minerals include predominantly spinel, magnesioferrite, magnesiochromite, and uvospinel minals (Table S1).

The mineral composition shows the "titanomagnetite trend" in the orangeite and lamproite fields [33] with increasing Fe and Ti and decreasing Mg, Al, and Cr concentrations. The $Fe_t^{2+}/(Fe_t^{2+} + Mg)$ ratio varies from 0.35 to 1 (Figure 3). The mineral composition of the spinel group minerals is consistent with the UML rock composition [5], including the ultramafic lamprophyres of the Chuktukon complex [7] (Figure 3). The spinel-group minerals contain the multiphase melt inclusions and they occur as mineral inclusions in olivine phenocrysts (see Melt inclusion investigations).

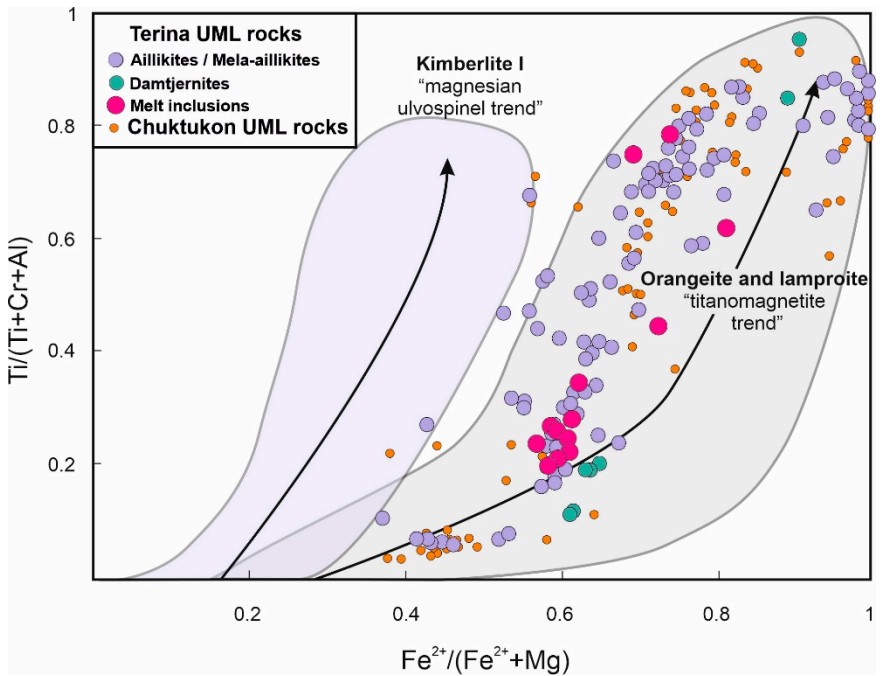

**Figure 3.** Atomic Ti/(Ti + Cr + Al) vs. $Fe^{2+}/(Fe^{2+} + Mg)$ plot for the spinel-group minerals of the Terina and Chuktukon [7] UML rocks. Kimberlite and orangeite trends are taken from [33].

*Perovskite* forms microgranular aggregates, subhedral and octahedral micro-crystals, and grains of 25–50 μm in the carbonate matrix and they are usually replaced by rutile (Figure 2c,d,g,h,m,n). The minerals include up to 0.3 wt% $Al_2O_3$, up to 0.17 wt% MgO, 1.16–1.71 wt% $Fe_2O_3$, up to 0.44 wt% $Na_2O$, up to 0.75 wt% $V_2O_3$, and up to 1.79 wt% $LREE_2O_3$ (Table S2). Secondary processes of the perovskite alteration cause a small anionic–cationic disequilibrium (Table S2). The perovskites from the Chadobets complex also contain a small amount of SrO (up to 0.41 wt%) [10].

The alteration of perovskite is presented by the microgranular aggregates of rutile, ilmenite, quartz, oxides, and hydroxides of Fe and Mn. The silicates of the zeolite group also replace the perovskite grains (Figure 2n). The zeolite-group minerals are closer in composition to natrolite, which occurs in association with hydrothermal minerals, such as chlorite, carbonate, serpentine, epidote, bastnäsite-(Ce), monazite-(Ce), and other secondary minerals.

The xenomorphic grains of *clinopyroxene* of the mela-aillikites and damtjernites are located in the groundmass of the UML rocks and they have an average size of 50–100 μm (Figure 2e–j,p). Mela-allikite contains up to 10–15 vol% and damtjernites include 5–10 vol% of clinopyroxene in the rocks. The mineral composition of clinopyroxene corresponds to diopside-hedenbergite varieties, with a presence of ferri-tschermacite and titano-tschermacite minals (Table S3). This composition is consistent with previous data for the Chadobets UML rocks [10]. At the same time, the zonal clinopyroxene crystals occur in the rocks, and the marginal parts of these crystals are similar to the composition of aegirine.

The evolution trend from diopside-rich clinopyroxene compositions towards the aegirine end-member is typical for many alkaline complexes in the world [34] (Figure 4). However, two opposite trends can be observed: from diopside to aegirine without the significant involvement of $Fe^{2+}$, for example, for the complexes of Katzenbuckel in SW Germany and Murun in Russia, or from diopside via hedenbergite to aegirine, for example, for Ilímaussaq in South Greenland. Moreover, there are many examples of intermediate trends in the evolution of clinopyroxene compositions between oxidative (ΔFMQ = +1..+2) and reducing (ΔFMQ = −2..−4) crystallization conditions, for example, in Lovozero, Russia, Alnö, Sweden), Coldwell nepheline syenites, Canada, and North Qôroq, South Greenland. The clinopyroxene compositions of the Terina UML rocks show a position around the fayalite-magnetite-quartz (FMQ) buffer and show an intermediate trend for clinopyroxene evolution, for example, alnöites in Sweden, Coldwell nepheline syenites of Canada, and North Qôroq alkaline rocks in South Greenland (Figure 4).

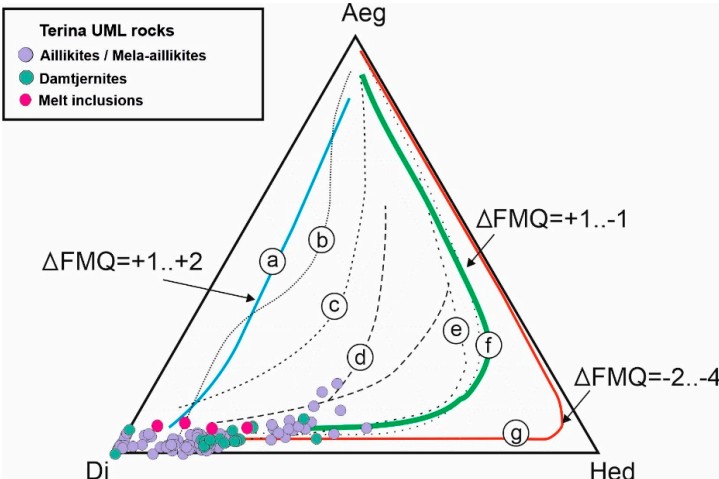

**Figure 4.** Clinopyroxene evolution trends of various alkaline complexes in the world according to [34]: (**a**) Katzenbuckel, SW Germany [34]; (**b**) Murun, Russia [35]; (**c**) Lovozero, Russia [36]; (**d**) Alnö, Sweden [37]; (**e**) Coldwell nepheline syenites, Canada [38]; (**f**) North Qôroq, South Greenland [39]; (**g**) Ilímaussaq, South Greenland [40]. Quantitative data on oxygen fugacity (given as ΔFMQ units, where FMQ is the fayalite-magnetite-quartz buffer) imply that the chemical evolution of clinopyroxene might be useful as a qualitative indicator of oxygen fugacity. The clinopyroxene compositions of the Terina UML rocks show the position around the FMQ buffer.

The primary *carbonates* of the UML rocks are presented by calcite and dolomite fine-grained matrix aggregates (Figure 2, Table S4). The quartz-dolomitic globules form the secondary hydrothermal aggregates, crystallizing with the filling of voids from the walls of voids to the center of the globules (Figure 2k,l). The mineral composition of the late dolomitic globules differs from that of primary carbonates by the higher $FeO_t$ content (6.82–11.98 wt% of $FeO_t$; Table S4). The groundmass damtjernite carbonates are dolomite and latter siderite (Table S4). The carbonates of the Terina UML rocks are Sr-bearing and they contain up to 2.25 wt% SrO (Table S4). The mineral composition of carbonates is similar to that of the carbonates (calcite and dolomite) of the Chuktukon complex [7].

*Phlogopite* forms euhedral macrocrysts and groundmass crystals and it has a zonal structure (Figures 2j and 5). The size of the macrocrysts ranges from 2 to 5 mm to the first cm; the groundmass crystals are 50–250 μm in size. The chemical composition of the macrocrysts and groundmass crystals is similar: 0.60–5.79 wt% of $TiO_2$ and up to 1.14 wt% of BaO (Table S5). Phlogopites have a low concentration of MnO (up to 0.46 wt%), and the F and $Cr_2O_3$ contents are below the detection limits (Table S5).

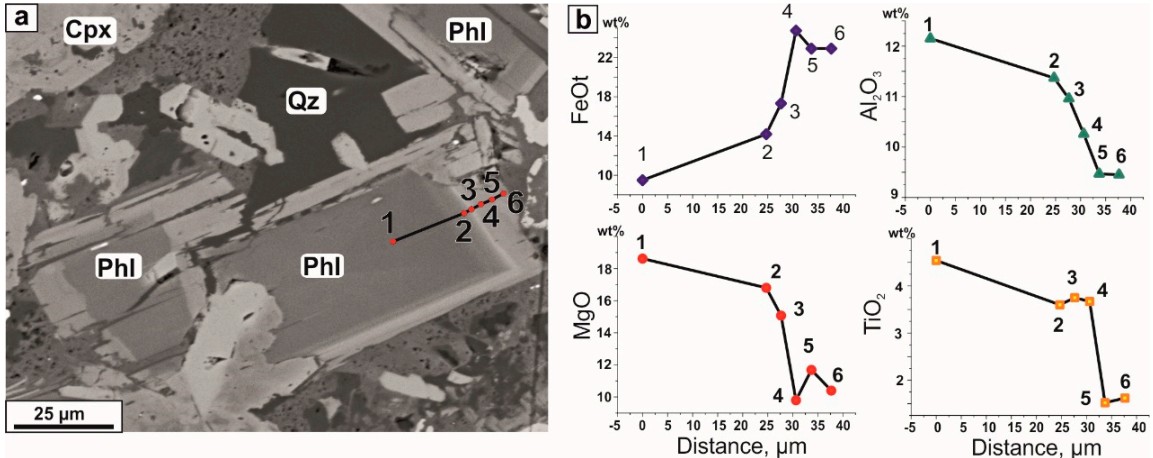

**Figure 5.** Mineral composition of phlogopite crystal according to electron microprobe analyses (EPMA) analyses (wt%): (**a**) BSE-photo, (**b**) variations of components along the profile line.

The profile of the chemical composition evolution of the phlogopite shows a rapid increase in the $FeO_t$ concentration as the MgO, $Al_2O_3$, and $TiO_2$ contents decrease from the central sites to the edge zones, and the Mg# decreases from 0.7 to 0.45 along the profile (Figure 5). The mineral composition of phlogopite shows the evolutionary characteristics of the magmatic crystallization processes. The trend of the FeO increase is similar to that of Chadobets phlogopites [7,10]. Epidote, quartz, calcite, rutile, and chlorite group minerals form in altered parts of the phlogopites.

Ba-containing phlogopites have been identified in UMLs of the Chuktukon complex [7]. Moreover, the high Ba content in micas is typical for many alkaline rocks, such as kimberlites [33] and ultramafic rocks of the alkaline carbonatite complexes, for example, the olivinites of the Guli complex from the Maymecha–Kotuy alkaline province [41], as well as the metasomatized mantle xenoliths of eastern Antarctica [42]. The phlogopite compositional range for the Terina complex is similar to the compositional range from other worldwide UML-carbonatite occurrences, and follows the UML rock trend of the Chuktukon complex [5,7,33] (Figure 6).

*Fluorapatite* has been attributed to the minor and accessory minerals. It forms rare euhedral grains and prismatic crystals with size parameters from 10–25 to 150–200 μm (Figure 2g,h,j). The mineral contents are as follows: 0.41–1.21 wt% $FeO_t$, up to 2.04 wt% MgO, up to 0.72 wt% $Al_2O_3$, 1.44–1.73 wt% $SiO_2$, up to 2.91 wt% $Na_2O$, 0.96–1.74 wt% SrO, up to 0.95 wt% $SO_3$, up to 2.42 wt% $Ce_2O_3$, up to 0.61 wt% $ThO_2$, 1.18–2.12 wt% F, and up to 0.08 wt% Cl (Table S6). The chemical composition of fluorapatite is typical of the alkaline ultramafic carbonatite complexes (e.g., [43–49]). Fluorapatite crystals from damtjernites contain lots of crystal-fluid inclusions (see melt inclusion investigations). The secondary carbonates replace the minerals; some grains of the fluorapatite are surrounded by, or include, the micrograins of monazite-(Ce), which could be caused by the remobilization of REEs within apatite by means of the hydrothermal processes [45–47].

Damtjernites differ from aillikites by the presence of feldspar and/or alkali feldspar crystals (up to 10 vol%) in the groundmass [5] (Figure 2m–p). Table S7 provides the mineral compositions of plagioclase and K-feldspar of damtjernites from the Terina complex. Previous studies have also identified the presence of nepheline in the UML rocks [9,10].

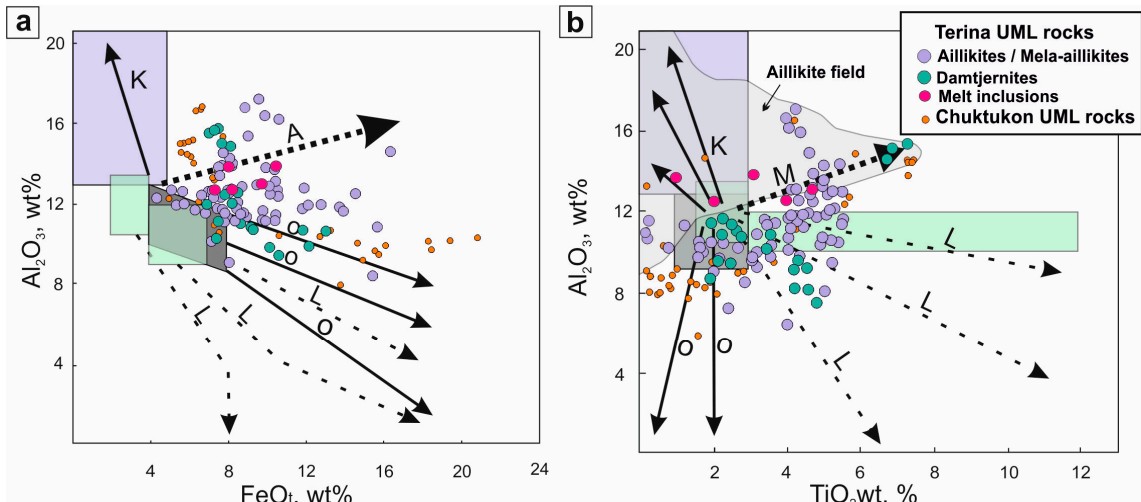

**Figure 6.** Compositional trends of micas from the Terina and Chuktukon complexes [7] UML rocks: K—kimberlite trends, O—orangeite trends, L—lamprophyre trends, M—menette trend [33]. The aillikite field is presented according to [6]. Diagrams of the variation in $Al_2O_3$ vs. $FeO_t$ (**a**) and $TiO_2$ (**b**).

*Albite* forms idiomorphic plates in the groundmass that are sometimes oriented in a certain direction (Figure 2m,o). The average size of albite crystals is 200–300 μm, and this can occasionally reach up to 0.5–1 mm. The mineral composition of plagioclase contains almost no anorthite molecule and a small amount of $K_2O$—up to 0.61 wt% (Table S7).

*K-feldspar* forms anhedral grains in the groundmass of up to 20 μm in size (Figure 2p). The mineral composition of alkali feldspar is orthoclase ($Or_{85–100}$) with small amounts of CaO and $Na_2O$ (up to 1.4 wt% of $Na_2O$, Table S7). The BaO content of the Terina complex feldspar minerals is below the detection limits.

*Olivine* was originally present and only investigated in the aillikites (Figure 2a–d). The two main mineral populations were distinguished according to their size: (1) subrounded or subhedral macrocrysts (1–10 mm) and (2) euhedral or less subhedral fine-grained crystals (<1 mm) in the groundmass. This division is quite typical for lamprophyres and kimberlites ([50–55] and many others), and these populations were previously interpreted as xenocrysts (macrocrysts) and phenocrysts (fine-grained crystals). However, in many works, the inconsistency of such division has been proved and, following [53,54], to avoid the genetic connotations, we divided the olivines into olivine-I (macrocrysts) and olivine-II (fine-grained crystals or phenocrysts) (Table S8).

In olivine-I grains, central parts (core) and rims (marginal part) are observed (Figure 7). The composition of the center part of different grains varies largely, even within the same sample. Some of them are significantly enriched in iron (#Mg 82 ± 0.3 and 74.7 ± 0.1), as well as Mn (1306 ± 20 ppm and 2140 ± 15 ppm) and Zn (154 ± 10 ppm and 210 ± 18 ppm), and they have minimal quantities of Ni (1440 ± 30 ppm and 1370 ± 20 ppm), Ca (485 ± 20 ppm and 270 ± 15 ppm), and Cr (<50 ppm) (Figure 7).

Olivine-II is homogeneous in the BSE images, with no significant differences in the forsterite content (#Mg 84–86.6). However, two different trends (or fields) can be clearly distinguished according to the trace composition: (1) mainly central and (2) mainly marginal parts of olivine-II (Figure 8). The core trend is characterized by a weakly varying and relatively low Ca content (750–1030 ppm), a slightly negative slope in the #Mg vs. Cr and #Mg vs. Mn plots, and a slightly positive slope in the #Mg vs. Ni plot (Figure 8). For the rim-trend, significant variations are noted for 1580–2790 ppm Ni, 930–2095 ppm Ca, 1121–1535 Mn, and 70–270 Cr (Figure 8).

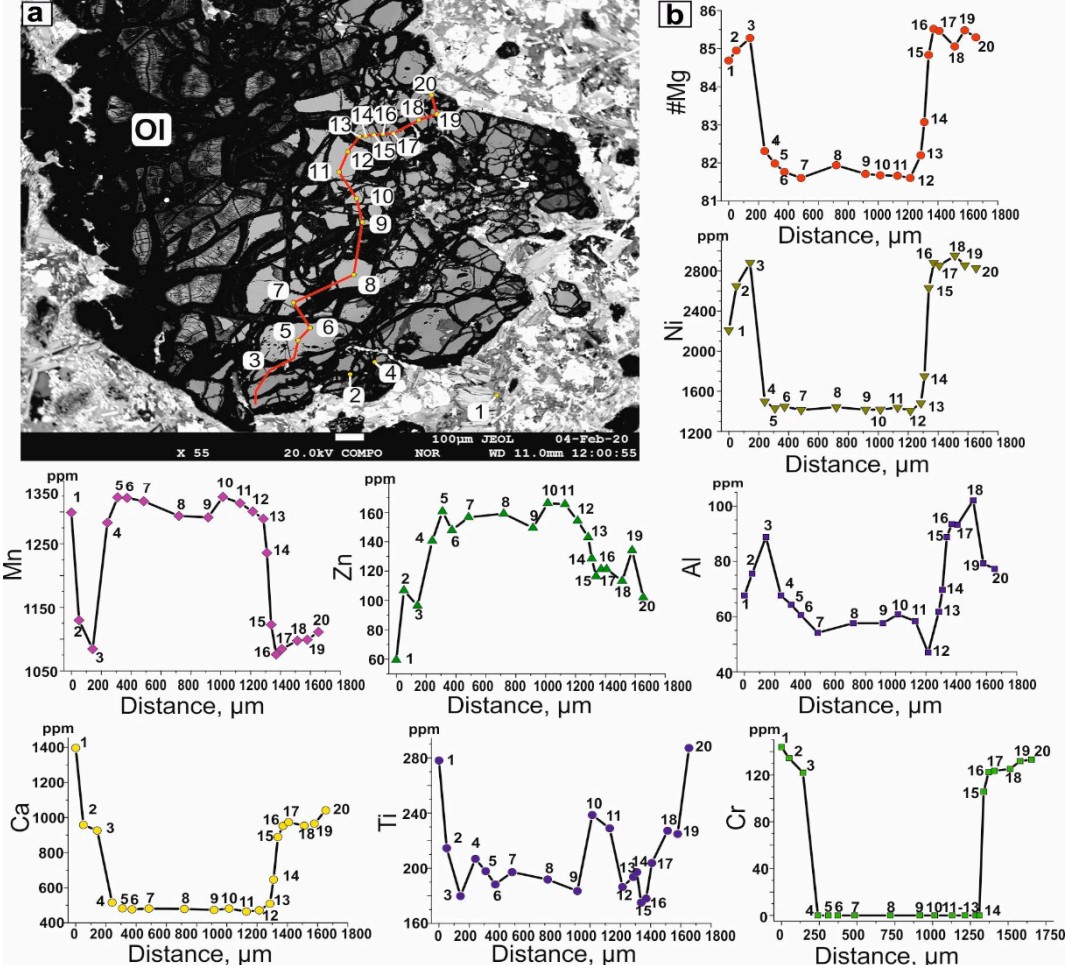

**Figure 7.** Variation in the trace composition of the olivine-I macrocryst with a relatively low #Mg core, according to WDS analyses (ppm): (**a**) BSE-image with profile line and (**b**) the variation in the components along the profile line.

Another cores of olivine-I, on the contrary of presented before, are enriched in magnesium (#Mg 89–86.5), Ni (3080–3175 ppm), and Cr (190–270 ppm), and they have the low content of Mn (950–1080 ppm) (Tables S8 and S11, Figure S1). The compositions of the marginal zones (rims) of olivine-I almost completely coincide with the compositions of the olivine-II: the inner part corresponds to the center of olivine-II, and the outer part corresponds to the rim (Figure 8; Table S8). Often, serpentine replaces the marginal parts of olivine-I (Figure 2a).

The obtained data are consistent with previous analytical data regarding the olivine composition in the Chadobets complex [9,28] (Figure 8). The present results complement the variation in the trace composition of the olivines of different populations within the Chadobets alkaline UML-carbonatite complex. In olivine-I, a few rounded ilmenite and orthopyroxene mineral inclusions, as well as polyphase melt inclusions, which form linear structures and appear to be secondary [56], were identified, while olivine-II contains spinel-group minerals, such as Cr-spinel and Ti-magnetite (see melt inclusion investigations).

Quartz, calcite, serpentine, dolomite, epidote, chlorite, rutile, barite, synchysite-(Ce), and monazite-(Ce) represent secondary minerals of the Terina complex UML rocks. The minerals are related to the hydrothermal-metasomatic stage of alteration of the ultramafic lamprophyres of the Terina complex. These mineral assembleges occur interstitially among the grains of primary minerals and/or form the networks of microveinlets.

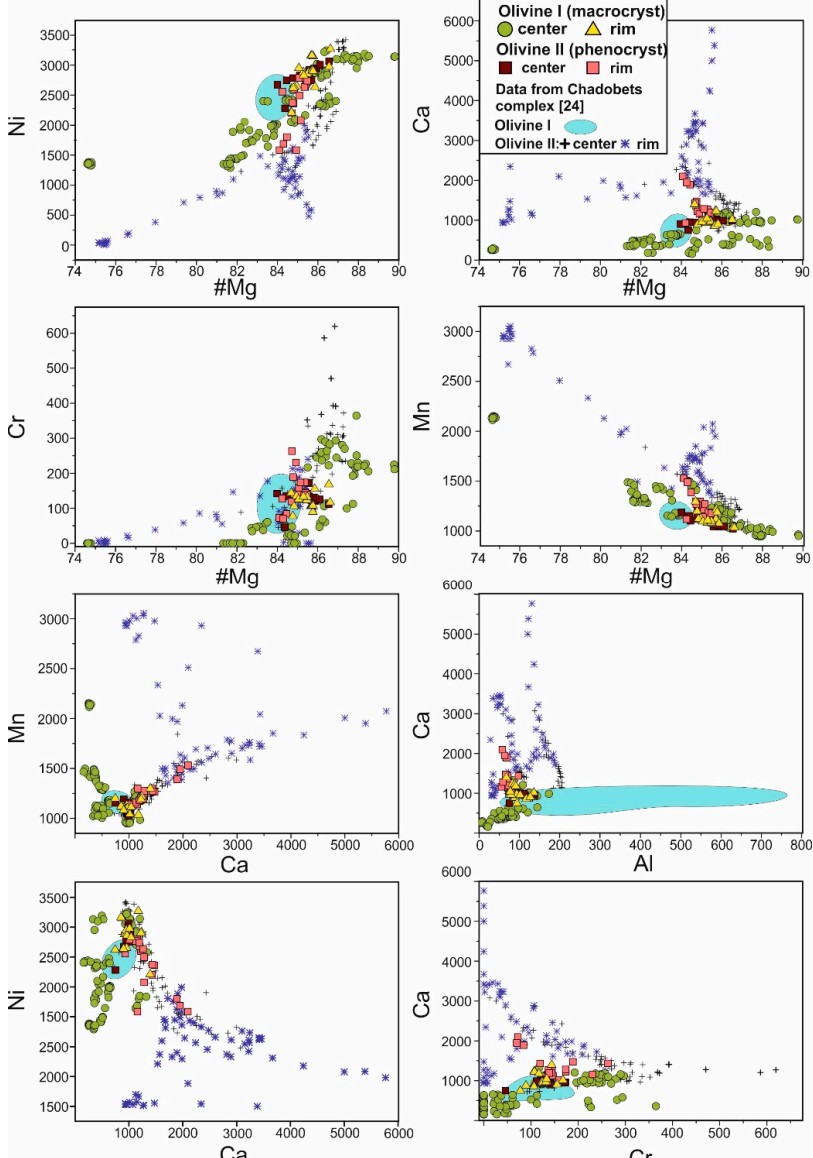

**Figure 8.** Diagrams of the variation in #Mg vs. Ni, Cr, Ca, and Mn, and Ca vs. Mn, Al, Ni, and Cr in olivine macrocrysts (olivine-I) and phenocrysts (olivine-II) of the Terina complex UML rocks, as determined by WDS analyses (ppm). The legend for the plots is shown as an insert in the second graph. The figure also contain the data for the Chadobets complex from [9].

Table S6 provides the REE mineral phase composition (Supplementary Materials). *Monazite-(Ce)* comprises 0.56–0.68 wt% $SiO_2$, up to 0.6 wt% $Al_2O_3$, up to 0.73 wt% $FeO_t$, 1.05–1.23 wt% CaO, up to 0.96 wt% $Na_2O$, 0.89–1.68 wt% SrO, 0.36–0.39 La/Ce, and 0.72–0.88 La/Nd. The genesis of monazite-(Ce) within the fluorapatite grains might be related to the remobilization of REEs within the apatite crystals due to the hydrothermal-metasomatic processes [45–47], as mentioned previously in the text. *Synchysite-(Ce)* includes 0.58–1.54 wt% $SiO_2$, up to 0.91 wt% $Al_2O_3$, 1.38–2.42 wt% $FeO_t$, 1.81–6.67 wt% CaO, 7.21–9.14 wt% CaO, 0.52–0.56 wt% La/Ce, and 1.68–1.92 wt% La/Nd (Table S6). *Barite* includes up to 2.74 wt% of SrO.

Rare micron impregnations of sulphides such as pyrite, chalcopyrite, pyrrhotite, and pentlandite (with solid solution decay), as well as ilmenite, rutile, and titanite, represent the minor and accessory minerals. *Ilmenite* contains up to 1.17 wt% $Nb_2O_3$ and up to 2.28 wt% ZrO, and *rutile* includes up to 0.85 wt% $V_2O_3$.

## 4.2. Melt and Mineral Inclusion Investigations

Melt and mineral inclusions were investigated in the olivine macrocrysts (Ol-I) and phenocrysts (Ol-II), as well as in the Cr-spinel crystals of the aillikites; the crystal fluid inclusions in the fluorapatite of the Terina complex damtjernites were also studied (Figures 9–12). These are the first data on the melt inclusion composition of the Chadobets UMLs. The previous melt and mineral inclusion investigations in zircon of the Chuktukon carbonatites revealed the presence of Na–K- and Ca–Ba–Sr–REE-carbonates and alkali silicate and fluorapatite mineral phases [8].

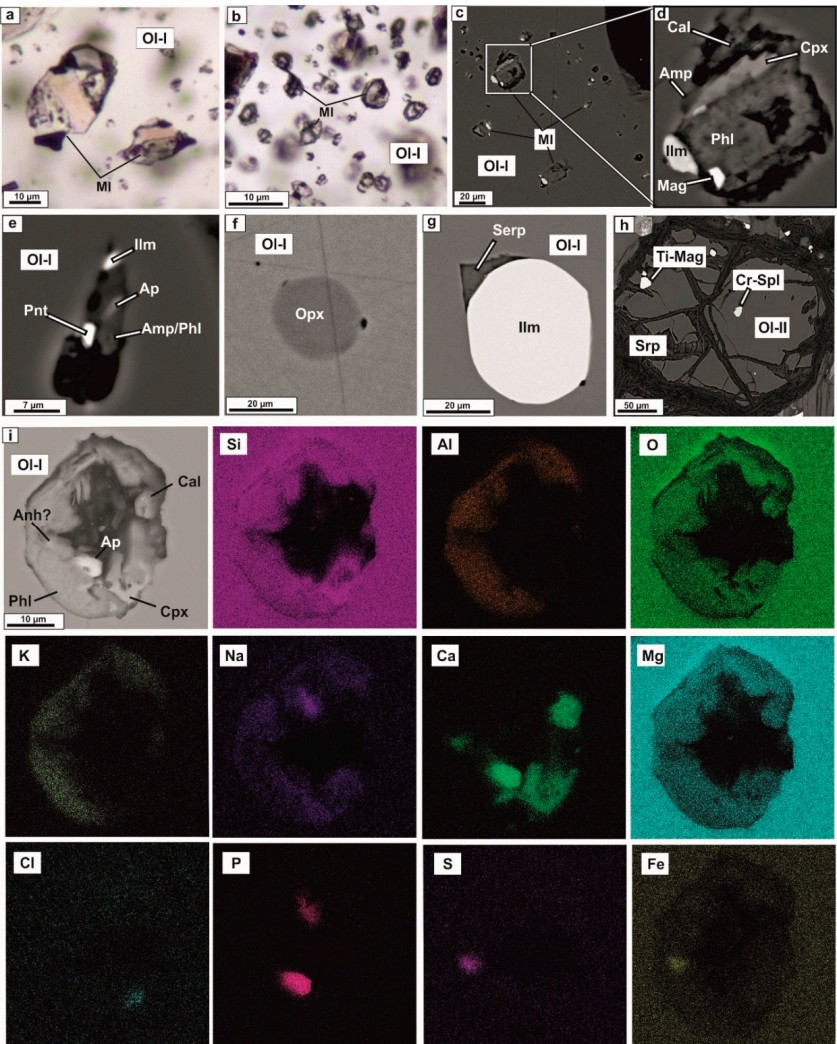

**Figure 9.** Melt inclusions (MI) in the olivine macrocrysts (Ol-I) and phenocrysts (Ol-II) of the aillikites of the Terina complex: (**a**,**b**) photomicrographs of the secondary melt inclusions under the optical microscope; (**c**–**e**) BSE photos of the opened melt inclusions which are composed of ilmenite (Ilm), magnetite (Mag), amphibole (Amp), calcite (Cal), clinopyroxene (Cpx), phlogopite (Phl), and apatite (Ap) daughter-phases and the xenogenic pentlandite (Pnt) crystal phase; (**f**) BSE-photo of the orthopyroxene (Opx) and (**g**) ilmenite mineral inclusions with serpentine (Serp) in the olivine macrocrysts; (**h**) spinel-group (Cr-Spl) and Ti-magnetite (Ti-Mag) mineral inclusions in the olivine phenocryst (Ol-II); and, (**i**) BSE-photo and electron microprobe (EMP) X-ray element maps showing the chemical composition of the opened melt inclusion in olivine-I: calcite (Cal), apatite, clinopyroxene, phlogopite, and probably anhydrite (Anh) are present as daughter phases.

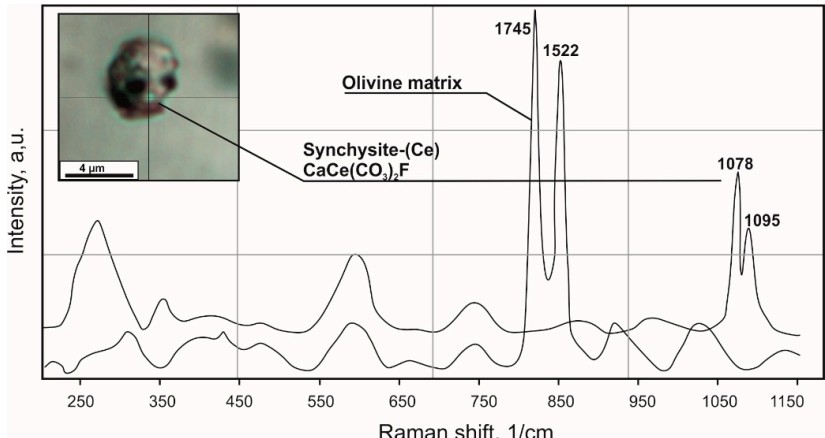

**Figure 10.** Raman spectroscopy analyses of the crystalline daughter phase of the secondary melt inclusion in the olivine macrocryst of the aillikites from the Terina complex.

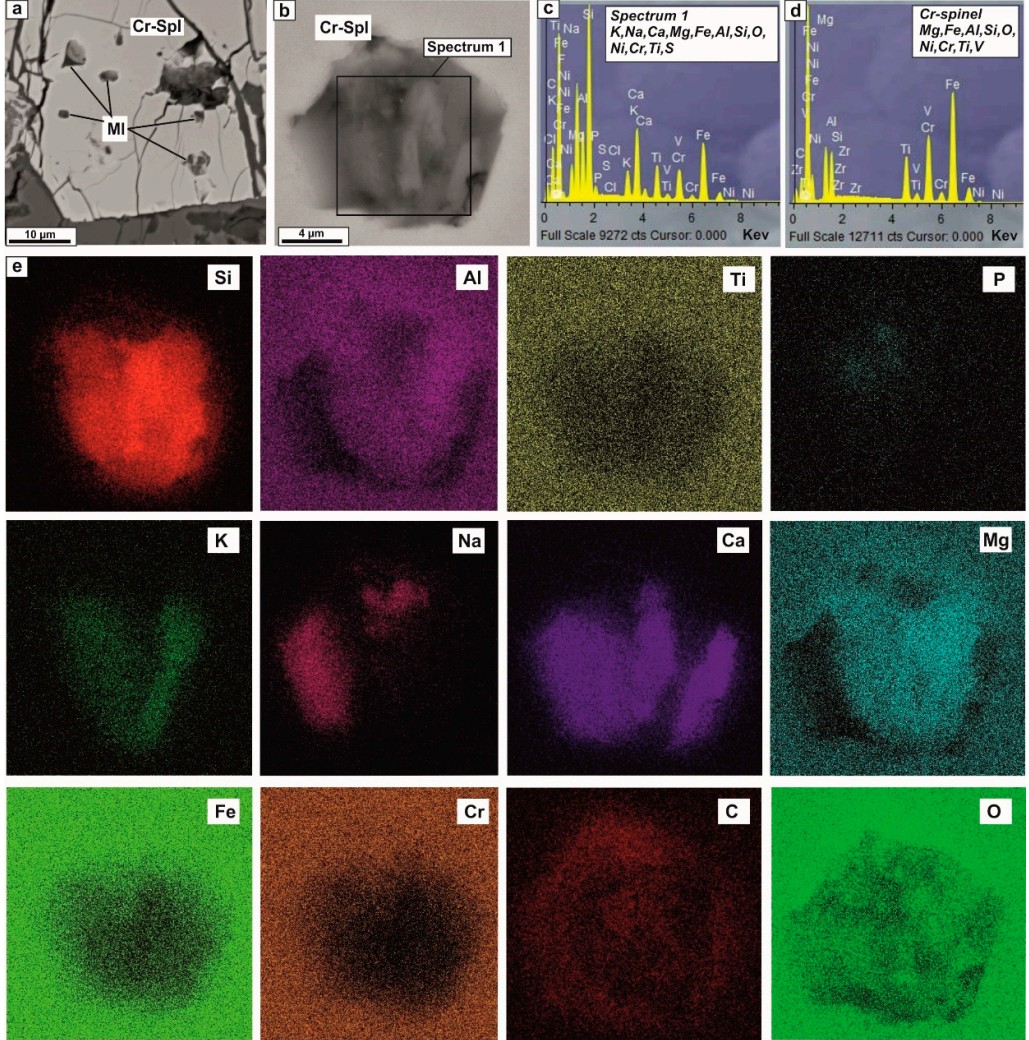

**Figure 11.** Melt inclusions in the Cr-spinel crystal from the aillikites of the Terina complex: (**a**) BSE photo of the locations of the inclusions; (**b**) BSE-photo of the opened melt inclusion; the SEM spectra of the inclusion (**c**) and Cr-spinel matrix (**d**) compositions; and, (**e**) the EMP X-ray element maps show the chemical composition of the opened melt inclusion.

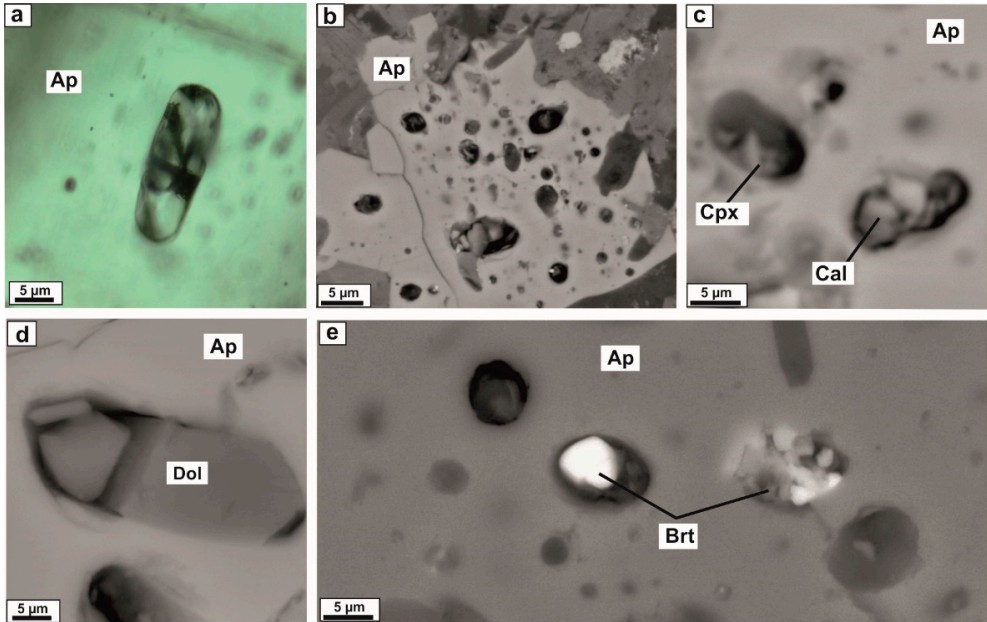

**Figure 12.** Crystal-fluid inclusions in the apatite of the damtjernites of the Terina complex: (**a**) microphotograph of the crystal-fluid inclusion; (**b**–**e**) BSE photos of the opened inclusions containing crystalline phases of clinopyroxene (Cpx), calcite (Cal), dolomite (Dol), and barite (Brt).

*Olivine macrocrysts* (Ol-I) contain melt and mineral inclusions (Figure 9a–g,i). The melt inclusions are located in the healed fractures and planes within the olivine crystals and they have been related to the secondary melt inclusions captured [56]. The inclusions have a round or elongated vacuole shape, ranging from 10 to 20 μm in size and consisting of several crystal phases (Figure 9).

Scanning electron microscope (SEM) analyses and electron microprobe (EMP) X-ray element mapping of the opened melt inclusion revealed the presence of clinopyroxene, phlogopite, Ca-amphibole, calcite, fluorapatite, ilmenite, and Ti-magnetite phases (Figure 9a–d, Table S9). The mineral composition of the crystalline phases for a single system of inclusions was preserved, which points to the homogeneity of the captured melt and allows for us to attribute these phases to daughter phases. Pentlandite (Figure 9e) and the S-containing mineral phase, probably anhydrite (?) (Figure 9i), were diagnosed in several opened inclusions. Raman spectroscopy analyses of the non-opened melt inclusions determined the presence of a synchysite-(Ce) [CaCe(CO$_3$)$_2$F] crystalline daughter phase in addition to the above described ones (Figure 10). We also do not exclude the presence of other water-soluble phases, for example, carbonates, sulfates, or chlorides as daughter phases of the melt inclusions, which could have been destroyed during the preparation of the open melt inclusion investigation by SEM and EMP.

The compositions of the spinel-group minerals (Figure 3), phlogopite (Figure 4), and clinopyroxene (Figure 6) crystal phases of the melt inclusions in olivine macrocrysts are similar to the mineral phase compositions of the aillikites from the Terina complex, according to the electron microprobe analyses (EPMA) (Tables S1, S3, S5 and S9). The clinopyroxene in melt inclusions is predominantly related to the diopside–hedenbergite series (Figure 4), and the phlogopite daughter-phase contains BaO (up to 0.61 wt%).

The olivine macrocrysts also contain the mineral phase inclusions of orthopyroxene and ilmenite (Figure 9f,g; Table S9). The size of the mineral inclusions ranges from 15 to 25 μm. The orthopyroxene mineral inclusions in the olivine macrocrysts are more similar to those of enstatite and they correspond to the formula Mg$_{1.71–1.72}$Fe$_{0.25–0.27}$Ca$_{0.01–0.03}$Si$_{1.97–1.98}$Al$_{0.01–0.02}$Ti$_{0–0.01}$O$_6$. The TiO$_2$ content in the orthopyroxene inclusion composition is up to 0.4 wt% (Table S9).

*Olivine phenocrysts* (Ol-II) include the Cr-spinel group and Ti-magnetite mineral inclusions (Figure 9h; Table S9). The crystalline inclusions have subhedral and euhedral forms with 10–20 μm

parameters. The inclusions are located in the olivine phenocrysts in the following way: the Cr-spinel group minerals are situated in the central parts and Ti-magnetite occurs in the marginal parts of the olivine crystals (Figure 9h).

The composition of the mineral phase inclusions in the olivine phenocrysts includes Cr-spinels with a high $TiO_2$ content (ulvospinel minal; Table S9) in the central parts of the olivines and Ti-magnetite crystals with a relatively higher $TiO_2$ concentration (up to 19.4 wt%) in the marginal parts (Figure 9h, Table S9). The compositions of the inclusions are similar to those of the mineral phases of the rocks (Figure 3).

According to [56,57], *spinel-group minerals* contain primary melt inclusions (Figure 11). The inclusions are polyphase, have round, negative crystal or octahedral forms with a size of 5–15 μm (Figure 11a,b). Unfortunately, their small size made it difficult to determine the mineral composition of the inclusions; however, the EMP X-ray element maps of the opened melt inclusions show that the crystalline phases are predominantly silicates, so it can be assumed that one of the phases is phlogopite. In addition to Al and Si, the following elements were also identified in the opened melt inclusions: K, Na, P, Ti, Ca, Mg, and C (Figure 11e). The SEM analyses also revealed the presence of S in the melt inclusion composition (Figure 11c,d). These data are consistent with the results of the secondary melt inclusions investigations in the olivine macrocrysts.

*Fluorapatites* in the damtjernites of the Terina complex are filled with crystal-fluid inclusions (Figure 12). The apatite grains are xenomorphic relative to the main magmatic mineral assemblages and most likely crystallized at the late stages of the magmatic crystallization processes (Figure 12b). The size of the crystal fluid inclusions is 5–15 μm. The inclusions have round, elongated, or irregular shape vacuoles and different phase compositions (Figure 12). The crystal-fluid inclusions are located in the central parts of the apatite grains and they can be classified as primary trapped in heterophase systems [56,57]. SEM analyses of the mineral composition of the opened crystal-fluid inclusions in the fluorapatite were conducted to establish the presence of clinopyroxene, calcite, dolomite, and barite crystalline phases (Figure 12b–e).

## 5. Discussion

### 5.1. Genesis of the UMLs of the Chadobets Complex

The mineral assemblages of the Terina UMLs are consistent with previous mineralogical data for the Chadobets complex [7,9,10,25,28,29,58] and they agree with the petrogenesis of ultramafic lamprophyres around the world ([1,2,5,6,33,35,36,59] etc.). We summarized our data on the mineral and melt inclusion investigations of the Terina UML rocks with the petrology of the Chuktukon UMLs and carbonatites [7,11,12,28–30,58].

The first phase of the Terina complex contains aillikites and mela-aillikites that were identified in the present paper, as well as in the Chuktukon complex [7,28]. The phases diagnosed in the melt inclusions in olivine and their compositions correspond to the mineral phases from the studied UMLs (Figures 3, 4 and 6). This allows for us to conclude that the melt in the healed cracks of the olivine macrocrysts is the parental melt for the aillikites of the Terina complex. The melt inclusion investigations in the spinel group minerals confirm the composition of the UML parental melt. Our melt inclusion investigations, together with the mineralogy of the UMLs of the Terina complex, show that the silicate-carbonate parental melt of the aillikites is significantly enriched in CaO and MgO (clinopyroxene, calcite, Ca-amphibole, and fluorapatite daughter and mineral phases), $TiO_2$ (ilmenite inclusions and daughter-phases, perovskite mineral), $K_2O$ (phlogopite daughter and mineral phases), $Cr_2O_3$ (spinel-group minerals), NiO (olivine and pentlandite mineral phases), $Al_2O_3$ and $SiO_2$ (alumino-silicate minerals and daughter phases), FeO (magnetite), $P_2O_5$ (apatite daughter phases and minerals), F, Cl, and REE (synchysite-(Ce), phlogopite and fluorapatite), and $SO_3$ (anhydrite). In addition, the UML melt contains relatively small amounts of $Na_2O$ and BaO (phlogopite and clinopyroxene daughter phases and minerals) (Figure 13).

The origin of UML-carbonatite melts is usually related to the partial melting of a carbonated peridotite (e.g., [59,60]). Petrological investigations of UML of the Chadobets complex revealed that these rocks are the product of parental melts derived from a moderately depleted mantle (e.g., positive $\varepsilon$Nd) [7,10]. The high #Mg, Cr, Ni of the UML rocks [7,9,10], as well as the same characteristics obtained for the olivine, compositions of clinopyroxene and phlogopite suggest that the UMLs of the Terina complex crystallized from peridotite mantle-derived melts and have not undergone significant fractional crystallization. The presence of magmatic carbonates and the carbonatites on the Chadobets complex confirm that the mantle source of the Chadobets aillikites likely contains carbonatitic metasomatic agent. In addition, as for the Chuktukon UMLs [7], the presence of BaO content in mica in the Terina UMLs suggests that the parental melts for the Chadobets complex were derived from a carbonated source [41,42].

| Complex/ Type of Alkaline Rocks | Host mineral | | |
|---|---|---|---|
| | Olivine | Apatite | Zircon |
| | Melt inclusion crystal phases | Crystal-fluid | Mineral inclusions |
| **Chadobets comlex** Aillikites - mela-aillikites | Cal, Phl, Cpx, Ca-Na-Amp, Fe-Ti-oxides (Ilm, Ti-Mag), F-Ap and Anh (?), Syn (Carb-REE) Mineral inclusions: Opx, Cr-spl, Ti-Mag | | |
| Damtjernites | | Cal, Dol, Cpx, Ba-Sr-sulpates | |
| Carbonatites | | | Na-K- and Ba-Sr-REE carbonates |
| **Udachaya-East (Siberian Craton)** Kimberlites | Phl, Humite-Clinohumite, Monticellite, Djerfisherite, Mag, Olivine, Chlorides, Sulphates, Phosphates | | |
| **Mark (Canada)** Kimberlites | Ca–Mg- and K–Na–Ba–Sr- carbonates, Phl and Monticellite, Fe–Mg–Al–Ti oxides, K–Na-chlorides, Phosphates, Sulphides | | |
| **Bultfontein (Kimberley, South Africa)** MARID xenoliths | Na, K-carbonates, Phl , K-richterite, F-Ap, Fe–Ti-oxides, Ba-Sr-sulphates | | |
| **Transdanubian (Hungary)** Xenoliths of Lamprophyre dykes | | Predominantly Carbonates and Minor Sulphates of Ba and Sr | |
| **Tomtor complex (Siberian Craton)** Lamprophyres | Na-silicates K-silicates, K-feldspar, Carbonates (Mainly Calcite) | | |
| **Pian di Celle Volcano (Umbrian Kamafugite Province, Italy)** Melilitites | Silicate-Carbonate Melt → Ca-carbonate Liquids with High Ba, Sr, F, and Cl | | |

**Figure 13.** Melt and mineral inclusions composition of UMLs and carbonatites [8] of the Chadobets complex; kimberlites of the Udachnaya-East pipe (Siberian Craton) [53,61–64], the Mark kimberlites (Canada) [65], and MARID xenoliths of the Bultfontein kimberlites (Kimberley, South Africa) [66]; melt inclusion compositions of the mantle xenoliths of lamprophyre dykes from the Transdanubian (Central Range, Hungary) [67], the lamprophyres of the Tomtor alkaline-carbonatite complex (Siberian Craton, Anabar Shield) [68], and the melilitites of the Pian di Celle Volcano (Umbrian Kamafugite Province, Italy) [69].

The origin of the olivine macrocryst in the UML rocks is controversial and debatable. Tappe et al. [6] argued that macrocrysts could be phenocrysts. The compositions of the rims of the olivine macrocrysts from the Terina complex almost completely coincide with the compositions of the olivine phenocrysts. The olivine macrocryst cores of the Chadobets UML rocks have #Mg 84 ± 0.2 [9], and the authors explained the origin of the Fe-rich olivine macrocrysts as the evolving batch of ultramafic-alkaline melts, which crystallized at lithospheric mantle depths and were entrained into a new portion of aillikite melts to form the olivine xenocrysts, according to previous work. Our data for the olivine macrocrysts cores of the Terina rocks show a relatively wide range of #Mg and trace composition (Figure 8). The olivine macrocrysts with lower #Mg (75), Cr, and Ni could be xenogenic and could have been captured during the rising of the aillikite melts through the lithosphere mantle. The compositions of the olivine with #Mg 83–85 are consistent with the previous data [9] (Figure 8), and they may refer to macrocrysts that were formed as a product of the evolving UML melts. Additionally, the olivine macrocrysts with higher #Mg (up to 89) and high Ni have clear mantle characteristics and they probably represent mantle xenocrysts [70] (Table S11, Figure S1).

The primary UML melts for the Chadobets UML-carbonatite complex were derived from low-degree melting of carbonated peridotite mantle, and the initial melts probably transported the olivine macrocrysts from the lithospheric mantle (~150–180 km) to the intermediate magmatic chamber, located at a depth of 4–8 km, as mentioned above [7,25]. The emplacement of the carbonatites was previously explained as being due to silicate-carbonate immiscibility [7,25,28,71]. We did not find any evidence of liquid immiscibility in the melt inclusions of the Chuktukon carbonatites [8]. However, geological observations describe the presence of carbonate globules (up to 2 cm in size) in the silicate matrix of the Terina ultramafic rocks, which have been interpreted as immiscibility records [71]. The investigations of ocelli in the Chadobets lamprophyres clarified the late-stage evolution of the lamprophyre fluid–melt system [72]. Our detailed mineralogical investigations revealed the presence of quartz-dolomitic globules in the Terina aillikites, which are interpreted as cavity-filling hydrothermal products (Figure 2i). In addition, the study of the chemical composition of the melt inclusions in the aillikites and mela-aillikites of the Terina complex did not establish any signs of silicate-carbonate immiscibility in the UML rocks.

Intrusion of the damtjernite magmas is the third phase of the Chadobets alkaline complex formation. The rate of magma emplacement in the intrusions was very, as revealed by the presence of mantle xenoliths of peridotites and eclogites and by the shapes of the intrusions (diatremes and pipes), which are full of pelletal lapilli in the rocks, according to the mineralogical and geological data [7,25,28,32,72]. The data point to a separate magmatic pulse involving the introduction of ultramafic lamprophyres from beneath the intermediate chamber, which is supported by the similar geochemical characteristics of the damtjernites and the aillikites [7]. The demtjernites contain clinopyroxene (diopside with aegirine minal), Ba–Sr sulphates, and Ca–Mg–Fe carbonates (calcite and dolomite) as the crystalline phases (Figure 13). The composition of the crystal-fluid inclusions in fluorapatite from the damtjernites of the Terina complex shows the accumulation of Na and $SO_3$ in the orthomagmatic fluids.

After the intrusions, the all-magmatic suites of the Chadobets UML-carbonatite complex were subjected to intense chemical weathering processes, leading to the formation of a thick ore-bearing weathering crust in the Cretaceous warm and humid climate [7,28–31,59,72].

*5.2. Comparison of Inclusion Data with Well-Known in the World*

The data on the melt inclusion composition in the olivine of aillikites from the Terina complex could be compared with the data on melt inclusions in olivine of kimberlites (Figure 13). For example, melt inclusion investigations in olivine of the Udachnaya-East kimberlite pipe (Daldyn-Alakit province, Siberian Craton) showed the presence of daughter crystals of phlogopite, djerfisherite, magnetite, olivine, humite-clinohumite, monticellite, rare sulphates and phosphates, and of $CO_2$-rich bubbles [53,61–64]. Primary and pseudosecondary melt inclusions in olivine of the Mark kimberlite (Canada) include the following daughter phases: Ca–Mg- and K–Na–Ba–Sr-bearing carbonates,

K–Na chlorides, along with subordinate silicates (phlogopite and monticellite), Fe–Mg–Al–Ti oxides (periclase and perovskite), phosphates, and sulphides [65]. The pseudosecondary inclusions can also contain tetraferriphlogopite, kalsilite, and sulphates. Based on the melt inclusion composition, the authors suggest that olivine in the Mark kimberlite was crystallized from, and transported by, a variably differentiated, silica-poor, halogen-bearing, alkali-dolomitic melt, with an estimated minimum entrapment pressure of ~200–450 MPa (or ~6–15 km) [65]. The main difference between melt inclusions in the aillikites and kimberlites is the predominance of carbonate-chloride compositions in the parental melts for the kimberlites.

The mineralogical and melt inclusion characteristics of the studied UML rocks of the Terina complex are apparently similar to those of orangeites (or the Group-II kimberlites) and lamprophyres. Orangeites are $H_2O$- and $CO_2$-rich peralkaline ultrapotassic igneous rocks [33], which are now recognized as a distinct magma type, based on their specific mineralogy, mineral chemistry, bulk-rock major and trace element concentrations, and isotopic compositions [73–76]. For example, the melting of MARID (mica-amphibole-rutile-ilmenite-diopside) rocks alone fails to explain some of the geochemical features of orangeites, such as the high Mg# (~85) and Cr and Ni concentrations (~2.000 and 1.000 ppm, respectively) [77]. The minimum depth of origin of orangeite magma is 150–200 km, based on the thermobarometry of the entrained xenoliths, which corresponds to the lower part of the lithospheric mantle in the garnet stability field [66]. MARID xenoliths from the Bultfontein kimberlite (Kimberley, South Africa) contain primary carbonate-rich inclusions in clinopyroxene with alkali (Na, K) carbonates, fluorapatite, abundant phlogopite (and tetraferriphlogopite), K-richterite, and Fe–Ti oxides and minor strontian barite as daughter phases [66] (Figure 13). The melt inclusion composition is similar with our data for melt inclusions in olivine, especially the presence of phlogopite, fluorapatite, carbonate, and Fe-Ti-oxides daughter-phases.

Apatite and K-feldspar from the clinopyroxene-rich mantle xenoliths of lamprophyre dykes (Transdanubian Central Range, Hungary) host melt inclusions with predominantly carbonate and minor S-bearing daughter crystals, such as sulphates of Ba and Sr [67] (Figure 13). The data revealed that the clinopyroxene–apatite–K-feldspar–phlogopite assemblage was formed by carbonatite metasomatism of an ultramafic mantle source, and the apatite and K-feldspar trapped the melt inclusions at ~1120 °C [67]. The study of primary and secondary melt inclusions in olivine, kaersutite, apatite, and titanite phenocrysts of the Tomtor complex lamprophyres (Anabar Shield, Siberian Platform) revealed the presence of sodic-silicate and potassic-silicate with K-feldspar and carbonate (mainly calcite) inclusions in the rocks [68]. It was determined that the minerals of the Tomtor lamprophyres formed at 1150–1090 °C from the sodic melts and at 1200–1250 °C from the potassic ones [68]. In contrast, the carbonate melt inclusions in the olivine and melilite of the Pian di Celle Volcano (Umbrian Kamafugite Province, Italy) consist of silicate–carbonate and predominantly carbonate varieties, which characterize the silicate–carbonate immiscibility processes in melilitites, and the phenomenon that seems to have first occurred at significant depths and temperatures above 1300 °C [69]. The salt melts evolved from silicate-bearing alkaline to predominantly calcic carbonate liquids with a simultaneous enrichment in Ba, Sr, F, and Cl [69].

The data presented for the orangeites and lamprophyres are complimentary with the melt and crystal-fluid inclusion compositions of the UMLs of the Terina complex and they could be used to estimate the chemical composition, as well as the PTX-parameters of formation of the aillikite melt.

## 5.3. PTX Estimations for the Chadobets UMLs

Nosova et al. [9] determined the olivine equilibration temperatures and pressures based on the composition of the olivine macrocrysts from the Terina UML rocks. The estimated temperatures were ~1000 °C at 20 kbar and 1200 °C at 60 kbar. We calculated the olivine equilibration temperatures for the macrocryst olivine cores that we analyzed while using the Al-in-olivine thermometer [78]. The calculation was carried out for the pressure of 2 and 6 GPa; olivine macrocrysts with mantle characteristics were used to determine the temperatures: high Ni (≥2350 ppm), low Ca (≤715 ppm),

and low Mn (≤1160 ppm) [70] (Table S11, and Figure S1 with BSE image of the high-Mg# olivine-I). The results were ~1020 °C at 20 kbar and ~1300 °C at 60 kbar, and the data are consistent with the previous temperature estimations for the Ilbokich olivine phenocrysts of 1240–1340 °C [79]. The estimations are consistent with the observations reported for an asthenosphere-lithosphere boundary beneath the Chadobets upland at a depth of 150–180 km [26]. In addition, the calculated temperatures agree with the Siberian pre-trap and post-trap mantle paleogeotherms [80].

The oxygen fugacity ($f$O$_2$) for the Terina ultramafic lamprophyres was calculated while using equation [81] with a temperature of 1020–1300 °C and pressure of 2–6 GPa (according to our calculations), and using the compositions of olivine II (Table S8) and spinel cores from groundmass (Table S1) and inclusions (Table S9). The calculated Δlog($f$O$_2$) FMQ values were near the FMQ buffer and they ranged from −0.06 to −0.22 and from −1.02 to −1.15 log units at 1020 °C and 1300 °C, respectively (Table S10). The qualitative estimation of the oxygen fugacity obtained from xenomorphic pyroxene (Figure 4) and its comparison with the estimated parameters for the olivine–spinel pair suggest that the oxygen fugacity increased slightly during fractional crystallization.

A biotite-apatite geothermometer [82] was applied in order to estimate the temperature of crystallization of fluorapatite and phlogopite in the Terina UMLs. There are no substantial differences between the macrocrysts and the central part of the groundmass phlogopite composition, while the rim of the latter contains zones with a sharp increase in the iron (FeO$_t$) concentration caused by the fractional crystallization processes, according to the EPMA analyses. Accordingly, for the solidus temperature estimation, the parameters of the macrocrysts and the core of the groundmass phlogopite (Table S5) with the fluorapatite composition data (Table S6) were applied. The results show that the original solidus temperature for the couple minerals ranges from 770 to 730 °C, and the latter crystallization processes have an interval of 680–575 °C (Table S5).

## 6. Conclusions

1. The mineral composition of the aillikites, mela-aillikites, and damtjernites of the Terina complex predominantly consists of olivine macrocrysts and phenocrysts, as well as phlogopite phenocrysts in a carbonated groundmass containing phlogopite, clinopyroxene (for mela-aillikites), and alkali feldspar (for damtjernites). Some olivine macrocrysts are likely to be xenogenic and they were probably captured during the rising of the aillikite melts. The olivine macrocryst rims and the olivine phenocrysts are the products of the evolving UML melts. The compositions of clinopyroxene, phlogopite, and spinel-group minerals of the Terina complex lie in the compositional range reported for other worldwide UML-carbonatite occurrences.

2. The composition of melt inclusions in the olivine macrocrysts and in the Cr-spinel crystals, together with the mineralogy of the UMLs, shows the PTX-characteristics of the parental silicate-carbonate melts for the ultramafic lamprophyres of the Terina complex. The estimated temperatures for the macrocryst olivine cores are 1020 °C at 20 kbar and 1300 °C at 60 kbar. The calculated oxygen fugacity ($f$O$_2$) values for the olivine–spinel pair are near the FMQ buffer and the clinopyroxene mineral characteristics show that the parameter slightly increased during the fractional crystallization of UMLs.

Our data suggest that the UML rocks of the Terina complex were crystallized from peridotite mantle-derived melts and did not undergo significant fractional crystallization. The primary melts for the Chadobets alkaline-carbonatite complex were derived from a carbonate-rich source, which was the product of mantle phlogopite-carbonate metasomatism beneath the Siberian Craton, described for the Chadobets complex in recent works [7,10].

**Supplementary Materials:** The following are available online at http://www.mdpi.com/2075-163X/10/5/419/s1, Table S1: Representative electron-microprobe analyses of spinel-group minerals from aillikites and damtjernites of the Terina complex; Table S2: Representative electron-microprobe analyses of perovskites from the Terina complex. Table S3: Representative electron-microprobe analyses of clinopyroxene from the aillikites and damtjernites of the Terina complex; Table S4: Representative electron-microprobe analyses of carbonates from the Terina complex; Table S5: Representative electron-microprobe analyses of phlogopite from aillikites and damtjernites

from the Terina complex; Table S6: Representative electron-microprobe analyses of fluorapatite, monazite-(Ce), and synchysite-(Ce) from the Terina complex; Table S7: Representative electron-microprobe analyses of feldspars from the damtjernites of the Terina complex; Table S8: Representative electron-microprobe analyses of olivine from the aillikites and damtjernites of the Terina complex; Table S9: Representative EPMA analyses of the mineral inclusions and the crystalline daughter phases in the melt inclusions in olivine of the aillikites from the Terina complex; Table S10: The oxygen fugacity (*f*O2) for the Terina rocks calculated using equation [80]; Table S11: The olivine macrocrysts equilibration temperatures for the Terina rocks calculated using Al-in-olivine thermometer [69]; Figure S1: BSE image of the high-Mg# olivine-I macrocryst from the Terina complex.

**Author Contributions:** Conceptualization, I.P. and A.S.; Data curation, A.D.; Investigation, I.P., A.S., A.D., Y.N. and V.P.; Methodology, I.P., A.S. and A.D.; Project administration, I.P.; Writing—original draft, I.P.; Writing—review and editing, A.S. and A.D. All authors have read and agreed to the published version of the manuscript.

**Funding:** The investigations were supported by the Russian Science Foundation (RSF), project #19-77-10004.

**Acknowledgments:** The authors express their sincere appreciation to the editors and anonymous reviewers for their contributions to improving the manuscript. The analytical equipment used in this study was provided by the Analytical Center for multi-elemental and isotope research SB RAS, Novosibirsk, Russia. The work was done as a state assignment of IGM SB RAS (0330-2016-0002) and Geological Institute SB RAS (AAAA-A16-116122110027-2).

**Conflicts of Interest:** The authors declare no conflict of interest.

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
