# Peer review of "Petrogenesis of Ultramafic Lamprophyres from the Terina Complex (Chadobets Upland, Russia): Mineralogy and Melt Inclusion Composition"

_minerals, doi:10.3390/min10050419_

Round 1

Reviewer 1 Report

Review on manuscript minerals-789367 by Prokopyev et al.

"Petrogenesis of Ultramafic Lamprophyres from the Terina Massif (Chadobets Upland, Russia): Mineralogy and Melt Inclusion composition"

This manuscript presents new and original data for melt inclusion in ultramafic lamprophyres (UML) from Chadobets Upland, Russia. It is the first data for whist ye of inclusions from this rocks in this area. Also, the investigation of inclusions from UML’ minerals is rare and this work is a very important contribution to mineralogy and petrology of UMLs.

The manuscript contains detailed mineralogical data for rock-forming minerals as well as for inclusions and obtained data have a high scientific level. But the Discussion section should be restructured because Authors main conclusions on the UML’ origin do not base on the obtained data and based on previously obtained data for these rocks. First of all, in the discussion, Authors should emphasize the contribution of present investigations of inclusion composition to the origin of UMLs. Authors obtained very important and interesting data but discussed the previously obtained results (in general).

I recommend publication of the manuscript pending major revisions. I stich my comments below and added them (with some minor comments) in .pdf file. Finally, the English is not good in the term of style (the Discussion section) and this paper needs to be corrected by a native English speaker.

Best wishes,
22.04.2020

  1. Introduction setting

I think that the Introduction section should be focused on the main issues of a study and any geological data should be moved into special Geological section. So I believe that lines 57:67 could be moved into Geological setting. Also, because one of the study issues is determinate or estimate the parental melt composition it is better if the Introduction will contain main models described the origin of UML melts in general.

Lines 37:44 – I am not sure that the explanation of the aillikite term is necessary for the introduction section because nowadays aillikite and damtjernite are widely used terms and several research papers have used a lamprophyre terminology including papers regarding Chadobets rocks (cited in the present paper). I believe that the reference to Tappe et al 2005, 2006 at line 47 will be enough in the introduction section and lines 37:44 could be omitted.

Line 58 – Why did you use melilitites as part of UML? According to both classification scheme of IUGS and Tappe 2005 melilitites are rocks contained > 10 % of melilite. In this case, it could not be UML because UML contains < 10 % of melilite (Tappe et al., 2005 – M>90 % including carbonate). I think that it is more correct to list rock type as " It contains melilitites, ultramafic lamprophyres (aillikites, mela-aillikites and damtjernites) and carbonatites…”.

Lines 59:60 – Why you use “massif” term to describe the magmatic complexes within Chadobets upland? Usually to describe the magmatism of ultramafic alkaline rocks uses terms as field, cluster or magmatic complex. Also from the Introduction section, it is not clear why have rocks within Chadobets upland been separated into two different missives? If they have different ages, composition or smth else you should introduce it before using separation into two complexes.

Lines 61:62 – According to the referenced papers using Ar-Ar dating based on the composition of rippit is not obvious because the Ar-Ar system in this mineral is not investigated enough. I am sure that this data should be omitted from paper, because the age 231 ± 2.7 Ma is not complemented with the geological structure of the area, including data presented in this paper.

Line 67 – I think that here it is good to describe that just part of Siberian kimberlites has Triassic ages (commonly for kimberlites from northern parts of the Siberia Craton - Anabar and Olenek region). In this variant, it seems like all Siberian kimberlites are associated with Siberian Superplume that is not correct. Moreover, the Triassic kimberlites have Late Triassic (231–215 Ma) ages (Sun et al., 2014) that significantly different from Siberia traps ages (~250 Ma).

  1. Geological Setting

Lines 90:92 - Sea comments above: it should be detailed what is the geological and/or petrological background that allow determining two massifs?

Line 94 – According to Fig. 1, the south part of the Chadobets upland also contains rocks of the first stage.

Line 95 – Authors used “the Chadobets complex” and it is better than two massifs. Also, Authors used “the Chadobets complex” at line 97 and below in the text.

Line127 – it is a good place to describe the age of Chadobets UML here (data from the Introduction section).

  1. Results of investigations

Lines 178-179 – Authors provide the increasing of feldspar and/or alkali feldspar crystals up to 10 % in damtjernite. Is it about the third phase of the UML magmatism? Alternatively, is it about the UML within the first phase of UML?

This section contains just petrography of massive porphyritic rocks. What is about pipes and diatreme types of damtjernites (lines 105:106)? It is very interesting because the petrographical description of damtjernites formed pipes and diatreme are very rare.

Lines 286:287 - Figure 6 used data after Mitchell, R.H. Kimberlite, Orangeites and Related Rocks 1995. Original plots do not contain aillikite compositional trend (A) as shown in Figure 6. It is the trend of the mica from minette i.e. lamprophyres (not ultramafic) contained biotite and minor amphibole (hornblende). Therefore, it is not correct to call this trend “aillikite”. It seems better if Authors plots compositional trends of mica from UML Aillik Bay (data from Tappe et al., 2006) or data from Chadobets UMLs from Nosova et al., 2020. These studies contain compositional trends of mica from UMLs that could be used on these plots.

Line 371 - There are additional symbols that are not described in the legend (circles with different colours).

Lines 411:415 – Authors noted that clinopyroxene and phlogopite from inclusions could have high concentrations of TiO2. After that, Authors concluded “The phases diagnosed in the melt inclusions and their compositions correspond to the mineral phases from the studied UML rocks. This allows us to conclude that the melt in the healed cracks of the olivine macrocrysts is the parental melt for the aillikites of the Terina alkaline complex” (lines 415:418). High TiO2 contain with low Mg# values in minerals from UML or related rocks usually show the evolved nature of parental melts (for example papers of Tappe et al about UML). So, it seems controversial that melts that are parental to clinopyroxene and phlogopite with high TiO2 contain could be parental melts for rocks containing more primitive by composition minerals…

Lines 416:418 sound as conclusions and should be moved into the Discussion section.

Also, it will be good if Authors plot compositions of inclusion’ minerals on the plots with compositions of minerals from rocks (probably as supplementary data).

Discussion section

I think that the Discussion section should be restricted. The first subsection with the formation model very difficult for understanding because of based on the previous data not on the data from inclusions. I believe that the Discussion section should contain the discussion of the data obtained in the present research. For example, it will be logical if Authors discuss the PT conditions, after that discuss the important inclusion’ data and finally discuss the model.

Lines 467:471 repeat the geological information from the Geology section and should be omitted.

Line 473 and line 475 Authors used the term “lamprophyres” probably it is better to use UML.

Line 490 – probably UML melt?

Figure 13 – What is the geological position of melilitites? Are they only in Chuktukon massif?

Lines 503:515 – Poor English. Please rewrite the text and omitted the description of olivine zoning (that are results). Please keep just main idea about depleted mantle source contained carbonate-phlogopite metasomatic domains (zones, veins).

Lines 527:533 – The studied UMLs show important data for olivine composition that could provide a xenogenic origin for olivine cores with low Mg# values. However, Authors avoided discussion these data for olivine from Chadobets UMLs. I believe that the author’ interpretation of these data should be added here.

Line 547 – What is the Mg# number of these olivine macrocrysts from the lithospheric mantle? Usually, mantle-derived olivine (according to data for olivine from kimberlite and related rocks from worldwide occurrences) shows higher Mg# values than cores of studied macrocrysts. So, these suggestions look implausible. Otherwise, Authors need to add information about the composition of olivine from the mantle that compared with cores of studied macrocrysts and expand the idea with additional references.

Line 555 – Also silicate and carbonate globules in UMLs from Chadobets complex were described in Nosova, Sazonova, 2017 Ocelli in the early Triassic Chadobets ultramafic lamprophyre (SW Siberian craton): evidence of late-stage evolution of the lamprophyre fluid – melt system. Proceedings of XXXIV International Conference “Magmatism of the Earth and related strategic metal deposits” 4-9 August 2017, Miass, Russia. P. 168-171.

Figure 13. The model is difficult for understanding. The present figure has some differences from the original model by Doroshkevich et al., 2019 Lithos. On the model from Doroshkevich et al., 2019 damtjernites and aillikites present the different branches of mantle magmatism and they are originated from a common mantle source. The main difference that aillikite magmas experienced the differentiation/or liquidation in the intermediate chamber. In the present paper, the aillikite and carbonatite origin from the intermediate chamber from damtjernites magmas that it is not related to the phase sequence. I think that if Authors distinguished three phases it will be better to show three subfigures for each phase than the re-published previous model.

Line 592 – What is the type(s) of lamprophyres?

Lines 595:603 – This text should be rewritten in the next structure: (i) the nature of orangeite (melting of MARID-bearing lithospheric mantle); (ii) the depths of the generating melts; (iii) the composition of inclusions from orangeite or MARID xenoliths and its comparison with inclusion composition from Chadobets UMLs. In the present text, it is not clear why Authors describe inclusion within MARID minerals.

Line 613 - Tomtor

Subsection 5.2 contains good reviews of compositions of inclusions from different types of the alkaline rocks as well as their PT conditions. However, it needs some conclusions based on these reviews. These data are not discussed in this section or the conclusion section.

Lines 660:672 – Please note that Ilbokich UMLs have Devonian ages and the evolution of mantle source by time is discussing.

Lines 668:672 – What are the differences? Please add additional information and references.

Lines 666-672 – It is very difficult to understanding. Please rewrite this sentence. Also if all Triassic alkaline magmatism were related to Siberian plume, please note why in the different place of the Siberia craton the alkaline magmatism has variable composition from UMLs, lamproites to kimberlites.

Conclusion

Line 684 – What is lamprophyre type? Also, the compositional trend lies in Figure 6a between Author’ aillikite and orangeite trend (see others comments).

Lines 705:708 – There are not data (or good discussion them) to provide these conclusions. See the main comment.

Table S2 - Aillikite – mela-aillikite.

Reviewer 2 Report

The paper on the Permo-Triassic ultramafic alkaline rocks from the southern Siberia contains a wealth of mineralogical data that will be of interest to readers of Minerals. The results on melt inclusions are also very important; however, these data are not extensively used in Discussion. I recommend making a summary in a Table format of melt inclusion’s daughter minerals and, if possible, bulk compositions together with those literature data on kimberlites and carbonatites that are referred to in the paper. The Discussion is the least satisfactory part of the manuscript, especially the last part. What is the point in presenting “5.3. Relationship with the Siberian plume activity”, if the age coincides with the age of Siberian traps? I strongly suggest toning down all statements that relate to so-called Siberian superplume, because even if the heat was provided (which I personally doubt), the associated metasomatic event in the mantle should be considered seriously.

I made a lot of comments in the PDF file of the manuscript. There are also questions that the authors should either answer or delete the controversial statements. The paper will benefit from shortening (a lot of repetitions and unjustified/unsupported statements), better organization and presentation, and English editing by an expert in this field of research.

There are also several technical comments that the authors should attend to. In brief,

All figures should be presented in the same style. Please pay attention to the style and size of fonts when using text labels, the legend and scales. The first point of reference are Figures 5, 7, 8 and 10. A good example to copy is Fig. 4.

Reviewer 3 Report

The paper is a petrology classic with many petrographyc and mineralogical data. Indeed, it could be two papers given the amount of data and rocks described. I would suggest making the greatest effort to coagulate the data presentation section, which is long and demanding, and also would lead to a more detailed discussion that illustrates all your data well. The conclusions are disappointing, a lot of work and then a discursive and quite obvious conclusion. For me, the work is fine and there are no errors in the substance, it is now only for the authors' self-love to make it more effectively if they wish to follow my suggestions. As for the long bibliography, a series of recent papers on similar subjects have been excluded although relevant and it is strange because among the authors there are many colleagues of the same Russian nationality. All observations and suggestions for comparison with other specific bibliography are in the pdf annotated in the attachment

Reviewer 4 Report

This paper presents a large number of high quality major and trace element analyses for inclusions and minerals in ultramafic lamprophyres from the well - studied Chadobets complex (Siberia) in order to show the first data on the parental melt composition for these rocks based on melt inclusion studies and to understand the mantle metasomatism of the underlying lithosphere.

Despite numerous recently studies of the Chadobets lamprophyre mineralogy (e.g. Doroshkevich et al., 2019; Nosova et al., 2018), the paper provides new information of mineral composition and mineral inclusions.

The conclusions are largely supported by the data, although there are some unsupported statements.

An Introduction is poor problem-oriented.  

There are some questions regarding data presentation. Tables in Supplementary contain only representative mineral compositions, not all data. It is difficulty to verify the results, especially the TPfO2 calculation. The text replicates the tables in some cases.

Discussion section is an unsuccessful part of the article and the “5.1. Formation model for the Chadobets complex” is the most unsuccessful one. Discussion in 5.1 section mainly is not based on data obtained by authors in the present paper. There are many repetitions from previously published authors’ papers. Some statements are requiring clarification. The development of the discussion is not logical and linear: first authors discuss the formation model, and then they discuss PTX estimations. The presented model is very close to previously published in Doroshkevich et al., 2019. I would like the authors to provide more in-depth and logical discussion, based on the data obtained in present article.

Conclusions section contains repetitions of text fragments from above sections of the paper.

I present detailed comments below, in the hope that these will help improve the manuscript. I recommend major revision and I think that authors have real possibility to improve the manuscript in the time frame allowed.

Minor comments

Lines 54-56 please add a reference regarding the depths of primary melts and deep mantle carbon cycle under the Siberian craton

Lines 54-56. The reference [10 – Nosova et al., 2020] didn’t advocate that the Chadobets magmatism is coeval with the formation of the Siberian large igneous province

Line 91, hereinafter. Why do you use a term ‘massif’ for dyke swarms, sills, stocks and explosive pipes? I think the term ‘cluster’ would be more suitable (the Terina claster and Chuktukon cluster)

Line 172 please, add refer to [10]

Line 176 replace ‘matrix’ by ‘groundmass’. We use a term ‘matrix’ for an interstitial fine-grained clastic material in volcanoclastic kimberlite and UML and a term ‘groundmass’ to an interstitial melt solidification products in coherent (hypabyssal) ones (e.g. Scott Smith et al., 2013).

Lines 178-179 Why don't you mention nepheline in a damtjernite? There is abundant nepheline in the Chadobets damtjernite, see [10, 24].

Line 225 hereinafter replace ‘hydrothermal-metasomatic’ to ‘hydrothermal’

Lines 235-243 and Fig. 4 Plot of ‘Clinopyroxene evolution trends …’ is a well-known and widely used diagram. It is not necessary to refer an every trend source in the text. You didn't plot the every trend from the source yourself, did you? Please, refer to the article from which you take the plot along with the trends (Marks et al., 2001?; Mann et al., 2006 ?). The legend for trends should be given in the caption of fig 4.

Lines  256-257 globules are not ‘secondary  hydrothermal aggregates’. They are ocelli (globules) that have crystallized from fluid, which exsolved from lamprophyre melt at late stages. The globules from Chadobets UML have been described by Sazonova and Nosova, 2017 and Nosova et al., 2018 (already cited)

(Sazonova L.V., Nosova A.A.  Ocelli in the early Triassic Chadobets ultramafic lamprophyre (SW Siberian craton): evidence of late stage evolution of the lamprophyre fluid - melt system // “Magmatism of the Earth and related strategic metal deposits”. Proceedings of XXXIV International Conference. Miass, 4-9 August 2017. Moscow: GEOKHI RAS, 2017. С. 168–171).

Line 293 please, replace ‘ex.’ by ‘e.g.’

Line 304 please, replace ‘minal’ by ‘molecule’

Line 331 and hereinafter. I suggest  to replace ‘microcomponent’ to ‘trace’ and, please delete “Fo-minal”

Lines 337-338 I suggest to remove this phrase. The trace elements for olivine phenocrysts from the Chadobets aillikites have been studies by Nosova et al., 2018 (already cited), see Phenocryst composition section

Lines 324-325 please, add a BSE image of high-Mg# olivine-I (either to the text or to Supplementary S8). Such high-Mg# olivine had not described in the Chadobets rocks previously in contrast with Fe-enriched cores which detailed figure 7 is presented. I think that high-Mg# cores probably are xenocrysts based on low Ca and Mn (Table S8), see (Bussweiler et al. 2017, http://dx.doi.org/10.1016/j.lithos.2016.12.015)

Lines 339-342 You should confirm this by plotting the data from [24,28] at the Fig. 8.

Line 357 remove ‘wt% ‘ from La/Ce, and La/Nd

Line 374 remove over ‘the’ from ‘as well as the in the Cr-spinel crystals’

Lines 409-418 This would be more compelling if you could show that at the plots. You can create new plots to show similarity of minerals from inclusions and rocks.  

Lines 445-446 please, add a reference regarding the results of a previous study of secondary melt inclusions in the olivine macrocrysts.

Line 473 pathogenesis is typo - petrogenesis

Lines 483-491 Unclear why first you mentioned a silicate-carbonate melt, and then – a lamproitic melt. A lamproitic melt is not a silicate-carbonate melt, it is a silicate melt and the lamproitic melt is strange candidate to a parental melt of the aillikite (e.g. Tappe et al., 2006, already cited).

Your suggestion of the parental melt composition is exclusively qualitative and descriptive. It is not necessary to study inclusions that to list components of a lamprophyre melt.

I suggest to remove lines 482-491 to section 5.2 and add them to lines 574-614

Lines 507-508 Unclear: Ti concentration in olivine increase is related with Ti-magnetite crystallization?

Lines 502 and 512 not “primary” melt, it would be correct - “parental” melt

Lines 523-528 There is a repetition of lines 320-325. Please, remove

Lines 518-533  I haven’t see relations of olivine macrocryst and phenocryst discussion with the Chadobets formation model.

Line 557 not “Terna” – correct to “Terina”

Lines 552-557 You should confirm this by refer to Fig.2i and see comment to Lines  256-257

Lines 591-592 please, argued more detailed the similarity of inclusions from the studied UML and from orangeites

Lines 618-619 Please, add a table with olivine compositions (to Supplementary) that have been used to calculated temperature by the Al-in-olivine geothermometer and olivine–orthopyroxene–spinel oxybarometer. More detailed description of the T estimation is needed. There are two types of macrocryst: with Fe-rich cores and with high Mg# cores. What type (or both) is characterized by obtained temperatures? Are olivine compositions satisfied the criteria of the mantle olivine is suitable for Al-in-olivine geothermometer?

Line 626 not calculations, but suggestion. Is olivine composition used for the olivine–orthopyroxene–spinel oxybarometer of Ballhaus et al. (1991) in the calibration range (XFe(ol) < 0.15)? See comment to Lines 618-619.

Lines 628-631 refer to the fig. 4 where these clinopyroxenes are shown

Lines 643-647 Unclear paragraph. Why the widespread intraplate  compression and orogenesis at the margins of the Siberian craton at 250 Ma (??) was plume-related event? Compression and orogenesis are related to subduction. The phrase “plume-related event was produced by the thinning of the lithosphere…” – would correct to “plume-related event produced the thinning of the lithosphere..”

Lines 650-659 I suggest to remove this paragraph. You haven’t any new isotopic data to discussion; all data have been previously discussed in your paper [9].

Line 667 Please, remove the Ilbokich aillikites, they have Devonian age and are not related to the Siberian LIP (Kargin et al., 2016; Nosova et al., 2020, both already cited).

Fig. 1 Legend please improve an order: first -  Permian-Carboniferous .., than – Neoproterozoic…

Fig. 4 please, see comment to Lines 235-243

Fig. 6 There is calc-alkaline lamprophyre (minette) trend, not aillikite trend at the original diagram after Mitchell, 1995.

Fig 7 Caption replace SEM-photo to BSE-photo or BSE-image

Fig. 8 Plotting of olivine composition fields from [10] should increase usefulness of the figure for readers 

Fig. 13 I think that abundant mineral abbreviators prevent from understand the figure. They need to be reduced.

Supplementary see comment to Lines 618-619

24 04 2020

Round 2

Reviewer 1 Report

Prokopyev et al. have addressed most of the reviewers' comments so that the paper can be accepted for publication. The paper "Petrogenesis of Ultramafic Lamprophyres from the Terina Massif (Chadobets Upland, Russia): Mineralogy and Melt Inclusion composition" is a very good contribution deserving publication in Minerals.

Author Response

We thank to Reviewer 1 and highly appreciated the recommendations that significantly improved our manuscript!!

Best wishes,

Ilya P. and coauthors

Reviewer 4 Report

The authors took most of the comments into account. I think the article can be published now.

Author Response

We thank to Reviewer 4 and highly appreciated the recommendations that significantly improved our manuscript!!

Best wishes,

Ilya P. and coauthors